# Following news on social media boosts knowledge, belief accuracy and trust

Sacha Altay [1] ✉, Emma Hoes [1] & Magdalena Wojcieszak [2,3] ✉

Many worry that news on social media leaves people uninformed or even misinformed. Here we conducted a preregistered two-wave online field experiment in France and Germany ($N$ = 3,395) to estimate the effect of following the news on Instagram and WhatsApp. Participants were asked to follow two accounts for 2 weeks and activate the notifications. In the treatment condition, the accounts were those of news organizations, while in the control condition they covered cooking, cinema or art. The treatment enhanced current affairs knowledge, participants' ability to discern true from false news stories and awareness of true news stories, as well as trust in the news. The treatment had no significant effects on feelings of being informed, political efficacy, affective polarization and interest in news or politics. These results suggest that, while some forms of social media use are harmful, others are beneficial and can be leveraged to foster a well-informed society.

Over 5.07 billion people, or 62.6% of the world's population, use social media platforms for communication and information access. Social media have been accused of degrading the quality of public discourse by promoting low-quality content that is toxic, sensationalist, misleading and sometimes harmful[1]. Falsehoods spread fast on some platforms[2], algorithms recommend entertainment over news and public affairs[3], and negativity, outrage and content denigrating political opponents receive disproportionate attention online[1,4]. These trends may exacerbate polarization, radicalization[5], misinformation and disinformation-induced confusion[6,7], ultimately undermining democratic governance.

Scholars have developed numerous interventions to counter these trends. Most have focused on how to minimize or recognize 'bad' content on platforms (for example, clickbait, misinformative or radical) or how to improve critical thinking[5,8], with a focus on making people more sceptical of bad content[9]. Other scholars have tested various methods for helping people recognize manipulation techniques[10]. For example, encouraging people to think about accuracy reduces the sharing of unreliable news headlines[8] and fact-checking reduces belief in false claims[11]. These approaches, although effective, may have unintended consequences on the perception of 'good' content (that is, factual and verified news)[12–14]. A host of other interventions have been tested at the platform level, such as friction[15], downranking

like-minded sources[16], removing reshared content[17] or the introduction of chronological feeds[18]. Yet these interventions may be ineffective in that friction reduces sharing of reliable content[15] and chronological feeds increase exposure to untrustworthy, extreme and ideologically congruent sources[18–20], and can only be implemented by or in collaboration with social media platforms themselves[21].

Our approach is markedly different. Rather than focusing on minimizing 'bad' political content, we enhance users' exposure to 'good' content, namely factual and verified news and public affairs information that is already freely and widely available on platforms[22]. We incentivized social media users in France and Germany (total $N$ = 3,395) to follow politically balanced news outlets on Instagram and WhatsApp for 2 weeks. Our preregistered field experiment causally estimated the effects of following the news on social media in naturalistic settings on four crucial outcomes: (1) current affairs knowledge about events that occurred between waves, (2) awareness of circulating true and false claims, (3) belief accuracy (that is, discernment between true and false news stories about national and international political issues, such as the wars in Ukraine and Gaza or the European Parliament election) and (4) trust in news media organizations. In addition, we tested whether news consumption on social media impacts participants' feelings of being informed about politics and current events, interest in the news and politics and affective polarization (that is, participants'

[1]Political Science Department, University of Zurich, Zurich, Switzerland. [2]Department of Communication, University of California, Davis, Davis, CA, USA. [3]Center for Excellence in Social Science, University of Warsaw, Warsaw, Poland. ✉e-mail: sacha.yesilaltay@uzh.ch; mwojcieszak@ucdavis.edu

feelings towards political parties and countries, such as Palestine–Israel or Ukraine–Russia). We tested whether these effects depend on differential levels of compliance, measured using screenshots submitted by the users, as well as self-reported measures of compliance. Last, we used qualitative comments and post-test responses to assess participants' feedback on their experience and their intentions to continue following the recommended news outlets.

We expand past literature in three important ways. First, we build on the work on approaches that 'promote the Internet's potential to bolster rather than undermine democratic societies'[23]. Although it is feared that social media facilitate the spread of clickbait, misinformative and low-quality and harmful content, past efforts aimed at reducing exposure to such content often overlook the wealth of verified and factual public affairs information already freely available on platforms[24–26]. Despite its abundance, the vast majority of users do not consume much news or political information on platforms[27–32]. This low consumption of public affairs leads to uninformed citizenry, with various negative consequences for democracy[33–36]. By contrast, well-informed citizens are better able to discern between true and false news[37,38], more resistant to manipulation[39] and more likely to hold stable attitudes[40]. Recognizing this, recent work focuses on positive incentivization and encouraging social media users to consume news[25,26] to make citizens more resilient to informational threats[39]. For instance, Askari et al.[26] created bots that contextually replied to users tweeting about non-political topics and encouraged the users to follow verified and balanced news accounts. This intervention slightly enhanced the following and liking of news accounts. In a YouTube field experiment, Yu et al.[25] found that nudging the algorithm by playing videos from news channels in the background increases recommendations to and consumption of news over time and promotes more diverse news diets. Such positive incentivizations align with users' desire for accurate information and educational content, instead of divisive, hateful or false content[41].

Second, we offer a comprehensive causal test of the effects of following news on social media on current affairs knowledge, awareness of both true and false news stories, belief accuracy and trust in news media, outcomes crucial to a well-functioning democracy[33,42]. Researchers worry that news on social media has negative effects by increasing information overload and political polarization or by increasing the feeling of knowing, without actually learning anything[43,44]. A recent meta-analysis found no association between social media use and political learning in observational studies and substantively small increases in experimental work[45]. Most experiments, however, rely on forced exposure to news, which users may avoid in their daily lives, and test outcomes immediately after exposure. It is therefore not clear whether similar effects would emerge in naturalistic settings where users have full control over exposure, myriad content options and where the majority seeks entertainment[31,46–48]. The few field experiments that incentivized internet users to increase exposure to news in general[25,49] or to partisan news[50,51] found null or very limited effects on various outcomes, such as knowledge or polarization.

We build on these field experiments by also maximizing external validity, not forcing exposure, and measuring effects not immediately after (potential) exposure. Furthermore, we assess compliance in several ways (that is, requesting participants to submit screenshots showing that they followed the news accounts and enabled notifications from these accounts and asking them to list and later recognize these accounts), which allows us to systematically test the differential effects of different doses of on-platform exposure to and engagement with news.

Third, we examine these effects across two countries and two distinct, popular and understudied platforms that have been accused of facilitating the spread of harmful content[52]. In general, social media platforms have transformed the relationship between news media and audiences, such that news organizations create and disseminate news in line with platform logics and constrained by platform affordances, such as interactivity or algorithmic curation[53,54]. For instance, the use of platforms such as Twitter has been shown to increase political knowledge[55,56], probably because many used the platform for news and politics[4]. Yet the use of Instagram, WhatsApp or Facebook has shown to be mostly detrimental[37,57]. Possibly this is because news outlets publish 'softer' articles on these platforms[58] or because these platforms are not primarily used for news[22]. Despite the different features and distinct uses of various platforms, cross-platform evidence on the effect of news use is limited[53].

We focus on Instagram and WhatsApp, platforms with different affordances and distinct usage patterns. Instagram is a well-established image and video-based social media platform with over two billion monthly active users worldwide as of early 2024. WhatsApp is a messaging platform with numerous social media features (such as stories) and almost three billion unique active users worldwide[59]. Each offers distinct opportunities for incidental news exposure: Instagram provides the most opportunities as news content is displayed on the same page as friends' posts and stories, while WhatsApp offers fewer, as users must actively open the news tab to access news. As of September 2023, WhatsApp users can follow news channels for updates in a one-way broadcast. Crucially for our purposes, Instagram and WhatsApp are not typically used for news[22] despite news media organizations' presence on the platforms, making it worthwhile to systematically examine the effects of encouraging users to follow news accounts on these platforms. In particular, 65% of the German population uses WhatsApp and 31% uses Instagram, but less than one-third of the users report getting news on these platforms[22]. In France, 44% of the population uses WhatsApp and 31% uses Instagram, and between a third and a half of the users report getting news on these platforms[22]. Focusing on distinct platforms and countries minimizes the chances that the detected effects are attributable to any one context, an important advance over past work, which have mostly studied a single platform, typically Twitter or Facebook, in the USA.

Overall, we find that following news accounts on social media increases not only users' current affairs knowledge and their awareness of current events, but also enhances their belief accuracy (that is, discernment between true and false news) and trust in news media—without increasing feeling of being informed, affective polarization or interest in news and politics. These findings show the potential of putting 'good' information in users' online ecosystems, which disrupts informational silos on social media and provides a robust method to bolster democratic resilience. Our results also offer a positive perspective by challenging the widespread negativity about the impact of social media. Contrary to the idea that news on social media increases the feeling of being informed without informing[43,44], our study demonstrates that news on social media can in fact help people become better informed and contribute positively to the democratic process.

## Results

Additional details on the data and methodology are provided in Methods and Supplementary Information offers a detailed description of all materials and methods used in this study, as well as additional robustness checks.

We conducted a preregistered online field experiment in France and Germany (N = 3,395) to causally estimate the effect of following news on Instagram and WhatsApp for 2 weeks in ecological conditions. To take part in the study, participants had to use WhatsApp or Instagram and not already be following the news accounts included in the experiment. WhatsApp users were allocated to the WhatsApp branch while Instagram users were allocated to the Instagram branch. All users were later randomly assigned to the treatment or the control within their branch. Participants in the treatment were asked to follow two news accounts while participants in the control followed non-news accounts. Figure 1 details the flow of the experiment,

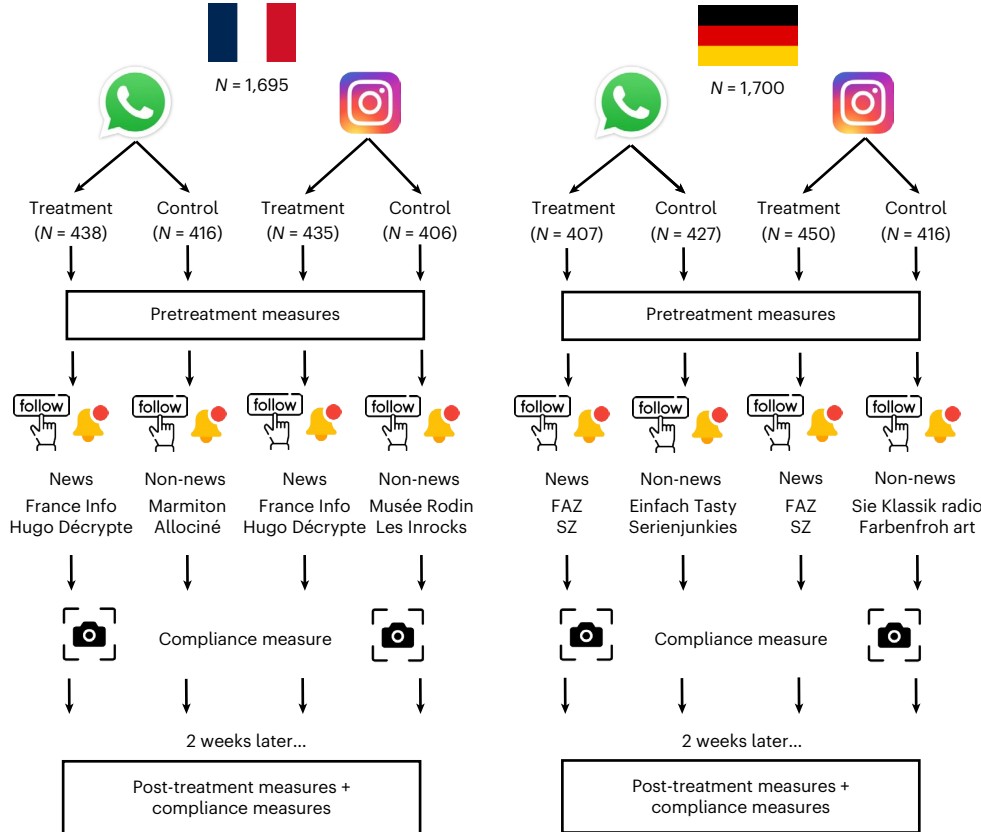

**Fig. 1 | Overview of the experimental design.** On the left side, the figure presents the experimental design for the French sample; on the right, the design for the German sample. WhatsApp and Instagram users were randomly assigned to the treatment or the control condition. Participants answered pre-treatment measures and were then asked to follow accounts on the respective platforms. At this stage, in wave 1 we measured compliance with the screenshots uploaded by the users. Two weeks later participants were surveyed again. In wave 2, we measured compliance with self-reported questions and participants answered post-treatment questions.

the number of participants in each branch and the social media accounts participants were asked to follow. The news media accounts were selected because they are generally trusted[22], reliable and not partisan (for more information see 'Treatment (wave 1)' section). The non-news accounts were selected because they do not cover news or politics and could be of interest to most people.

Below, we focus on the participants who completed both waves and uploaded at least a screenshot in wave 1 (regardless of whether the screenshot is valid). In Supplementary Information we report all exclusions. In total, 1,700 German participants (985 women, mean age of 40.5 years), and 1,695 French participants (1,042 women, mean age of 44.1 years) completed both waves. Age, gender and education were evenly distributed across control and treatment, and we observed no systematic bias in dropout across the conditions (Supplementary Information).

Our dependent variables were repeated across waves, and in wave 2 we added new political knowledge items and questions about new true and false news stories that appeared between the waves so that participants could not have possibly known them before the treatment. Supplementary Information details the wording of the questions for all the items used in the analyses and in Methods and Fig. 2 we offer an overview of our dependent variables. The use of dynamic dependent variables allowed us to maximize internal validity and assess over-time changes in current affairs knowledge, awareness and belief accuracy. Our field experiment also maximized external validity as we do not force exposure but instead incentivize it as part of participants' daily platform usage and measure effects up to 14 days after (potential) exposure.

We estimated the effects of our treatment using linear mixed-effect models with participants as a random effect, and control for age, gender, education and past vote (as well as country and platform), as preregistered. In particular, we modelled whether within-individual changes across waves are greater in the treatment compared with the control.

Figure 3 summarizes our main findings and breaks them down by country and by platform and Fig. 4 breaks them down by compliance levels.

First, the treatment led to greater gains in current affairs knowledge across waves, measured as the number of correct answers to the news quiz on national and international news. Current affairs knowledge increased among the treatment group by 0.12 points across waves compared with the control ($\beta$ = 0.07 (0.01 to 0.12), $P$ = 0.024). The gains are greater among participants who fully complied, with an increase in current affairs knowledge of 0.21 points ($\beta$ = 0.12 (0.05 to 0.36), $P$ = 0.008), and not significant among noncompliers.

Second, the treatments increased gains in awareness of true news stories (but not false news stories), namely the number of news stories participants reported having heard of or read before, while controlling for participants' tendency to report being aware of news stories they have not encountered with a placebo news story. The increase in awareness of true news stories across waves was 0.14 points higher in the treatment group compared with the control ($\beta$ = 0.08 (0.02 to 0.14), $P$ = 0.006). These gains are greater among participants who fully complied, with an increase in awareness of true news stories of 0.25 points ($\beta$ = 0.15 (0.06 to 0.24), $P$ = 0.001). The treatments did not significantly increase awareness of false news stories ($\beta$ = 0.03 (−0.03 to 0.10), $P$ = 0.32), regardless of levels of compliance or country platform-specific effects.

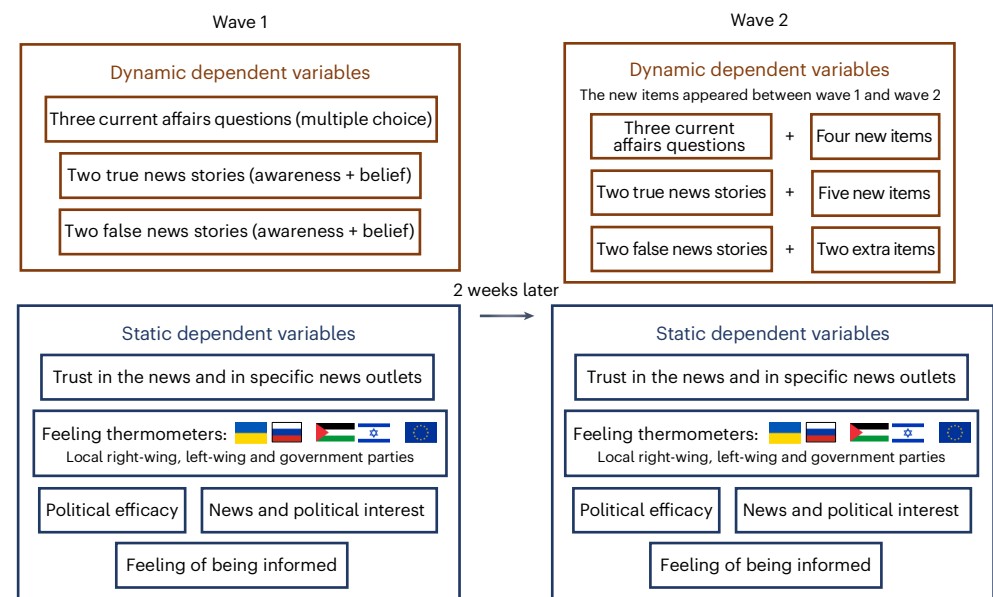

**Fig. 2 | Overview of the dependent variables.** On the left, the figure shows the dependent variables measured in wave 1, and on the right, those measured in wave 2. Dynamic dependent variables are highlighted in brown boxes, while static dependent variables are shown in blue boxes.

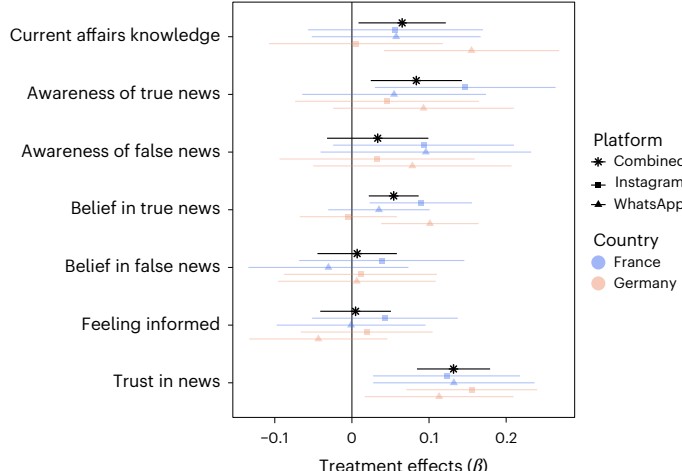

**Fig. 3 | Main treatment effects broken down by platforms and countries.** The black asterisks are the preregistered combined estimates across platforms and countries. The blue squares are estimates of the French Instagram treatment and the orange squares are the estimates of the German Instagram treatment. The blue triangles are estimates of the French WhatsApp treatment and the orange triangles are the estimates of the German WhatsApp treatment. The error bands represent the 95% confidence intervals. The $\beta$ represents standardized regression coefficients.

Third, we found substantial and statistically significant positive effects of our treatment on gains in belief accuracy across waves, computed as the difference between the perceived accuracy of true and false news stories[60]. Following two news accounts for 2 weeks increased belief accuracy by 0.70 points compared with the control ($\beta = 0.06$ (0.02 to 0.11), $P = 0.004$). These gains are larger among participants who fully complied, with an increase in belief accuracy of 1.04 points ($\beta = 0.10$ (0.03 to 0.16), $P = 0.004$), and nonsignificant among noncompliers. Overall, gains in belief accuracy are due to a higher acquisition of true beliefs across waves ($\beta = 0.05$ (0.02 to 0.09), $P = 0.001$) rather than to a lower acquisition of false beliefs ($\beta = 0.01$ (−0.04 to 0.06), $P = 0.80$).

Even though these results show that following news organizations on social media does increase current affairs knowledge and belief accuracy, participants in the treatments did not significantly feel more informed about current events or politics compared with controls ($\beta = 0.01$ (−0.04 to 0.05), $P = 0.84$)—none of the country platform-specific effects come close to statistical significance.

We also clearly show that across countries and platforms, the treatments increased trust in the news as well as trust in the specific news sources participants were asked to follow. Across waves, trust in news increased by 0.13 points on the six-point scale in the treatment compared with the control ($\beta = 0.13$ (0.08 to 0.18), $P < 0.001$). Regarding trust in the specific news outlets, in France, the treatment increased trust in France Info and Hugo Décrypte, while in Germany they increased trust in FAZ and SZ (Supplementary Fig. 1).

Finally, the treatments had no statistically significant effect on political efficacy ($\beta = -0.02$ (−0.07 to 0.03), $P = 0.46$), interest in the news ($\beta = -0.02$ (−0.09 to 0.04), $P = 0.46$), interest in politics ($\beta = 0.01$ (−0.04 to 0.07), $P = 0.61$) or on any of the eight feeling thermometers (Supplementary Information). The full results for these outcomes broken down by country and platform are reported in Supplementary Information.

### Robustness checks
We conducted a number of robustness checks and provide additional information in Supplementary Information. In sections 1 and 2 we report detailed results for the static dependent variables. In section 3 we break down the results by waves. In section 4 we report various measures of compliance. In section 5 we report qualitative comments from participants. In section 6 we offer the descriptives statistics of all dependent variables. In section 7 and on Open Science Framework (OSF), we report heterogeneous treatment effects. In section 8 we show no differential attrition and correct randomization. In section 9 we report all exclusions and screens. In section 10 and on OSF we report all the questions in the survey, while in section 11, we show that the results hold while controlling for news use.

### Discussion
Social media platforms have been accused of facilitating the spread of toxic, sensationalist, manipulative and otherwise harmful low-quality

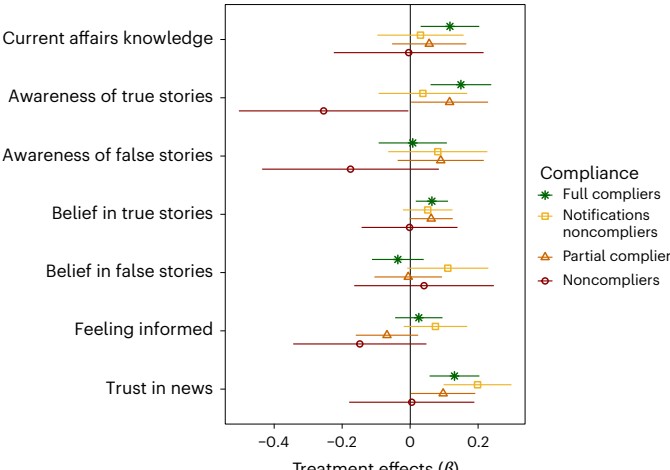

**Fig. 4 | Main results broken down by compliance levels.** Full compliers (*N* = 1,466) uploaded screenshots showing they followed the accounts and activated the notifications, and reported following the accounts in wave 2. Notifications noncompliers (*N* = 746) are similar to full compliers except that their screenshots do not show that they activated the notifications. Noncompliers (*N* = 240) did not upload a screenshot showing that they followed even one account. Partial compliers (*N* = 942) are the remaining participants; they uploaded screenshots showing they follow only one account and/or only activated the notifications for one account and/or reported having unfollowed the accounts in wave 2. The error bands represent the 95% confidence intervals. The *β* represents standardized regression coefficients.

content, which may foment polarization, increase misinformation endorsement and lower citizen support for democratic processes and institutions. Most interventions have tried to fight exposure to and spread of this 'bad' political content or teach individuals to recognize or resist such content. In contrast, our approach enhanced the 'good' content in users' social media ecosystem. As news media organizations have integrated platforms into their communication strategies, we tested whether incentivizing social media users to follow quality news on Instagram and WhatsApp can generate positive democratic outcomes.

Our field experiment on 3,395 WhatsApp or Instagram users in France and Germany offers three key findings. First, prompting social media users to follow news organizations increases current affairs knowledge, belief accuracy and awareness of true news stories, as well as trust in news, journalists and the news organizations followed. These effects are (probably) due to increased exposure to news content: participants in the treatment saw more news and learned from it. The effects on trust may be due to participants underestimating the quality of news coverage before the experiment and updating their perceptions after being exposed to this coverage, which in turn spilled over to perceptions of news and journalists more broadly. We note that these findings are more in line with past deactivation studies, which show that deactivating Facebook reduces knowledge[61–63], than with other, mostly cross-sectional, work, showing that platforms such as WhatsApp, Instagram or Facebook, have null or negative effects on political knowledge[25,45].

We emphasize that our findings emerged despite the naturalistic setting of daily social media use, which offers myriad distractions that could dilute the impact of the intervention. Exposure to news was not 'forced' and the treatment only consisted of following two news accounts for 2 weeks, which pales in comparison to the hundreds of other, potentially more interesting, accounts users follow. Even in the treatment group, news probably only represented a tiny portion of all the content participants saw on the platform. Still, some news content did manage to cut through the noise and produce measurable

beneficial effects. For these reasons, it is important to keep in mind that, despite being statistically significant, the treatment effects are small. We also found numerous nonsignificant effects, such as awareness of and belief in false news stories, the feeling of being informed, interest in news and politics, political efficacy and feelings towards Ukraine, Russia, Palestine, Israel, Europe and specific political parties. These null effects are in line with past work showing that attitudes are very resistant to change[16,18,25].

Second, the fact that these effects were detected in ecological settings and over time is particularly optimistic. Unlike fact-checking individual news stories or punctual accuracy prompts, our approach may build longer-term capacities and skills among users, helping them situate potentially problematic information in larger associative networks of factual knowledge gained from reliable media. Moreover, the gains in news trust of reliable outlets may translate into further gains in true beliefs, conditional on exposure to such content.

These two core findings together suggest that improving exposure to verified information from news media could be more effective than reducing exposure to low-quality information from junk news. The increased belief in true news and trust in news and journalists is important given the evidence that the rejection of true news may be a bigger problem than the acceptance of false news. For instance, people are sceptical of true news[64], distrust news media in general, including reliable news outlets[22] and trust mainstream news outlets much less than fact checkers do[65]. This suggests that improving exposure to reliable information may be more beneficial[26,66] than reducing exposure to misinformation, which—after all—represents a small portion of the news people see online[67].

Our third key finding regards the qualitative comments from the participants. These comments are overwhelmingly positive (Supplementary Information) and suggest that some users were largely unaware that it is easy and free to follow news organizations on Instagram and WhatsApp. This lack of awareness emerged especially in the comments from the WhatsApp sample, perhaps because the platform is mainly used for one-to-one communication and WhatsApp channels are a relatively new feature. Crucially, 51% of participants indicated that they intend to keep following these news organizations after the termination of the experiment. These qualitative insights suggest that the lack of awareness about the presence of quality news on platforms may partly contribute to the low levels of news consumption on social media (similar to the unawareness of censored information available via VPNs in China[68].

These comments add to the recent evidence that social media users report wanting to receive more educational and informative content[41] and that increasing recommendations and exposure to news on YouTube does not decrease users' engagement with and activity on the platform[25]. These comments, and our overall findings, suggest that following news generates positive effects, and calls into question the decisions of some platforms—most notably Meta—to minimize news dissemination. At the time of writing, news is downranked on Facebook, in some countries news outlets have been blocked from posting links to their content on Facebook and Instagram, and by default, the Instagram and Threads algorithms do not recommend news, users have to opt in. Meta claimed that people are not interested in news, that news is 'highly substitutable'[69], and that Meta 'never thought about news as a way to minimize misinformation/disinformation on our services'[70]. We suggest that there is a demand for free and reliable news on social media platforms not commonly used for news, and that, in line with past work[37,38], news can help people discern facts from fiction. People increasingly consume news on social media, and although they worry about inaccurate news, they find it convenient to use social media for news[22,71].

Although this project offers new optimistic evidence, we encourage future research to examine the generalizability of our findings.

For one, it is unclear whether similar effects would emerge on platforms with different features or uses. In our case, we do not detect statistically significant differences between platforms, but we lack statistical power to detect small differences. Yet there are reasons to expect that platforms with different algorithms and affordances would yield distinct results. For instance, if participants had followed news accounts on TikTok, they may never have been exposed to content from these accounts because users mostly use the 'For you' feed, which weakly prioritizes followed accounts. Similarly on YouTube, where the algorithms direct users away from news[3], subscribing to news channels may not increase news exposure in any pronounced ways (but tweaking the algorithm itself does generate these increases[25]). However, holding the quantity and quality of news constant, exposure to news on TikTok, YouTube or other platforms should also yield the benefits documented here.

Studies should replicate our findings in different countries and different times in electoral politics. Our experiment was run in two multiparty systems with strong public-service broadcasting and during relatively contentious times (with EU elections, the Israel-Gaza war and the Russian-Ukraine war). The effects may be different in other periods and contexts. Further, our approach is premised on the presence of reliable news outlets posting on social media, yet, in some countries or on some platforms, such outlets may be scarce. Moreover, our findings are contextual to the design of the platforms and their algorithms at the moment of the experiment[72], and so extending our work to different platforms, countries and time periods is needed.

Future work should also investigate the theoretical mechanisms behind the treatment effects. Breaking down the results by compliance levels offers some preliminary insights. The absence of statistically significant effects among participants who followed the news accounts but did not turn on the notifications (compared with participants who turned them on; the full compliers) suggests that notifications may have played an important role. Notifications could expose users to news content directly, cutting through the in-platform noise and informing users about current events and/or serve as a mere reminder for the users to access news. News notifications are not unusual. In wave 1, 47.5% of participants reported having received news notifications on their phones in the past week (in line with ref. 73).

The beneficial effects detected may be explained through cognitive processing theories[74]: the concise, visually-driven nature of short-form news on platforms such as Instagram and WhatsApp may enhance information retention by reducing cognitive overload, while the interactive features of these platforms could foster deeper engagement. Unlike traditional, longer formats, the processing of which requires deeper, more reflective processing (per the Elaboration Likelihood Model[75], short-form news may rely more on peripheral cues such as visuals or source credibility. This could mean that the cognitive and emotional pathways through which these formats influence knowledge retention, trust and content discernment differ substantially from those activated by traditional media. We encourage scholars to explore these understudied mechanisms.

We also encourage studies to assess the potential cumulative or over-time effects of our intervention[25,26]. If users engage with news on platforms, the algorithms are more likely to classify those users as politically interested, which increases future recommendations to news and public affairs[27,28]. Currently, there is a likely feedback loop where platforms deprioritize news, lowering users' exposure to news and leading users to seek out news less. Our approach could break these loops and set in motion a cycle where users receive updates from the news accounts they followed and from other accounts recommended by the algorithms[26]. On the downside, our approach could have negative substitution effects by increasing social media news consumption while reducing the consumption of higher-quality news. Yet such substitution effects are rare[18,76], rely on shaky assumptions and, importantly, are incompatible with our treatment effects.

Last, we add to the research on potential interventions to minimize information harms from social media platforms, yet we acknowledge that future research needs to examine whether and how our approach could be scalable. As an intervention, our approach could be implemented at scale either by platforms themselves, which could start prioritizing and recommending, not downranking, news in users' feeds, or by news organizations, which could increase their presence on social media and actively make the users more aware of this presence. Our findings clearly indicate that this could have democratically beneficial effects and—as the aforementioned research shows—would not minimize users' engagement with the platforms[25,41]. Alternatively, (targeted) social media ads, educational campaigns and informational outreach done by non-profits could be viable encouragement for some users to follow (more) factual and reliable news media organizations.

In summary, following news accounts on social media is a relatively rare behaviour among users, and accounts of celebrities or sports teams have disproportionately greater following on platforms[48]. Yet, as we show, users are willing to engage with news media when encouraged to do so, and this engagement increases current affairs knowledge, awareness of true news stories, belief accuracy and enhances trust in the news and journalists. This means that, with the right incentives, social media can be a powerful tool for promoting informed and engaged citizenship.

## Methods

This research project complied with all ethical regulations for research involving human subjects and received ethical approval from the University of Zürich PhF Ethics Committee (ethics approval no. 23.10.14). Participants provided informed consent at the beginning of each survey and were debriefed at the end of wave 2 (all the materials are available on OSF).

The preregistrations, materials, R scripts and the data to replicate the findings are publicly available on OSF.

### Participants

In 2024, between 1 and 11 March, we recruited 2,009 German and 2,021 French participants via the market research company Bilendi. Participants were recontacted between 18 and 28 March. In total, 1,700 German participants (985 women, mean age of 40.5 (12.5) years; median education of a 2-year college degree, 38% had a bachelor degree or more) and 1,695 French participants (1,042 women, mean age of 44.1 (11.5) years, median education of finished high school, 31% had a bachelor degree or more) completed the second wave. The median distance between waves was 14 days (mean of 13.4 days, s.d. of 1.74 days). In Supplementary Information we report all exclusions and show that randomization was successful and that there was no problem of differential attrition.

### Screening and randomization (wave 1)

Participants first consented to take part in the study and answered screening questions about (1) whether they have Instagram and WhatsApp accounts, (2) how frequently they use them and (3) whether they follow a list of social media accounts. To be eligible, participants had to have either an Instagram account or WhatsApp, use Instagram or WhatsApp, and not already be following the news accounts included in the experiment. After passing the screens, Instagram users were randomly assigned to the Instagram control or the Instagram treatment, WhatsApp users were randomly assigned to the WhatsApp control or the WhatsApp treatment, and those who used both Instagram and WhatsApp were randomly assigned to one of the four conditions (Fig. 1).

### Independent variables (wave 1)

After answering the screening questions, participants were told that they would have to follow two (at that moment, not yet specified) accounts on social media for 2 weeks and upload screenshots to show

that they complied. At this stage, participants were asked again if they consent to take part in the study.

Those who consented answered demographic questions (gender, age, education and vote at the last presidential election) and reported their news consumption frequency in general, on WhatsApp and on Instagram (from 'Never' to 'More than ten times a day'). Participants also reported their main source of news (for example, TV, social media and so on) and whether they received news notifications on their phones last week. TV was the main source of news for 37% of participants, followed by social media (22%), online (19%) and radio (17%). In wave 1, participants reported very low news consumption on WhatsApp and Instagram. In the WhatsApp groups, 62% reported never consuming news on WhatsApp and 15% reported consuming it less than once a day (mean of 1.87 on a six-point scale). In the Instagram groups, 29% of participants reported never consuming news on Instagram and 23% reported consuming it less than once a day (mean of 2.61). The questions on news consumption were taken or adapted from the Digital News report[22].

### Outcome variables (wave 1)

**Static outcomes.** In both waves, participants reported how informed they feel about the news and politics[77], how interested they are in the news and politics (from 'Not at all' (1) to 'Extremely' (6)[22], the extent to which they trust the news and journalists ('To what extent do you trust the media and journalists?' from 'Not at all' (1) to 'Completely' (7)), as well as the extent to which they trust the specific news sources included in the treatments (from 'Not at all' (1) to 'Completely' (7)[22]). We measured political efficacy with the following questions 'How well would you say you understand the important political issues facing our country?' (from 'Not at all' (1) to 'A lot' (5)) and 'Do you often think that politics and government are so complicated that you can't understand what's going on?' (reverse coded, from 'Never' (1) to 'Yes, most of the time' (5)). These questions are commonly used to measure internal political efficacy[78].

**Dynamic outcomes.** In wave 1, these outcomes are considered dynamic because they changed between the waves, such that new items were added in wave 2 (Fig. 2). All current affairs knowledge and news stories questions are reported in Supplementary Information.

We measured current affairs knowledge with three multiple-choice questions (with seven options, including 'Don't know') on the date of the next European election, the name of a French/German minister and the name of the southernmost town in the Gaza Strip. Such questions, in contrast with 'textbook knowledge' about how politics and governments function, are often used in surveys measuring political knowledge acquisition across numerous waves[37,55]. Our current affairs knowledge questions do not measure how well participants understand current political events, but rather their knowledge related to current affairs. Current affairs knowledge was computed as the sum of correct responses.

We used feeling thermometers (from 0 to 100) to measure how people felt towards Palestine, Israel, Ukraine, Russia and the European Union, as well as left-wing and right-wing political parties and the party of the current government.

We measured belief in, and awareness of, true and false news stories by asking participants to rate two true news stories selected from mainstream news outlets and two false news stories that have been fact checked as false by independent fact checkers. For each claim, participants were asked 'Before this survey, had you ever read or heard this statement?' ('Yes', 'No' or 'Don't know/Don't remember') and 'Do you think this statement is rather true or false?' (from 'Completely false' (1) to 'Completely true' (8))[37]. Awareness in claims was computed as the sum of 'Yes' responses.

### Treatment (wave 1)

All participants were asked to follow two accounts, activate the notifications and upload screenshots showing that they complied.

Participants in the WhatsApp conditions followed accounts on WhatsApp while participants in the Instagram conditions followed accounts on Instagram. Participants in the treatment conditions followed news accounts (France Info and Hugo Décrypte in France or FAZ and SZ in Germany), while participants in the control conditions followed non-news accounts about art (Musée Rodin or farbenfroh art), music (les Inrockuptibles or Sie Klassik radio) movies (Allociné or Serienjunkies) or food (Marmiton or Einfach Tasty).

While Hugo Décrypte is a news influencer (the most popular one by far in France[22]), his coverage is very similar to mainstream news outlets such as the ones included in the experiment. All the news accounts are popular on Instagram and enjoy a wide reach (followers count: Hugo Décrypte 3.9m, France Info 809k, SZ 811k and FAZ 631k), while they are markedly less influential on WhatsApp (Hugo Décrypte 406k, France Info 230k, SZ 36k and FAZ 16k). Regarding the formats, on Instagram the accounts use mostly videos and images, while Hugo Décrypte relies almost exclusively on images with text. On WhatsApp, France Info has an audio format with some text, Hugo Décrypte a text format with no links, while FAZ and SZ use a text format with links to (their) news stories.

### Treatment (wave 2)

Participants were first asked basic questions about their WhatsApp and Instagram use, and answered questions about their news consumption, their main source of news, as well as the static dependent variables (that is, trust in news, feeling thermometers, political efficacy, news and political interest and feeling of being informed).

Participants then answered the dynamic dependent variables. They answered the same three current affairs knowledge questions as in wave 1, plus four new current affairs knowledge questions about events that took place between wave 1 and wave 2. They rated the same two true news stories as in wave 1, plus five new news stories about events that took place between wave 1 and wave 2. They rated the same two false news stories as in wave 1, plus two additional false news stories about events that took place before wave 1 (we had fewer additional false than true questions because we could not find new false news stories circulating in France and Germany between waves). Participants also rated a 'placebo' news story that we made up ('Brazil has introduced a new tropical forest monitoring system that detects and prevents illegal deforestation in real time.'). We used responses to this placebo news story to control for participants' tendency to report being aware of news stories they have not encountered (as they could not have possibly encountered the placebo news stories before): 14.2% of participants reported having heard or read the placebo news stories before, while the median number of news stories participants reported being aware of was 4 (out of 11).

The dynamic dependent variables allow us to measure the acquisition of beliefs and knowledge, as well as gains in awareness. Such design is typically used in survey research[37,55] and reflects an interest in the change in the dependent variables (for example, beliefs and knowledge) rather than changes in the independent variables (for example, news consumption). Yet here, we combine this design with a field experiment in which we experimentally manipulate what is typically considered the independent variable (that is, news consumption), allowing us to draw stronger causal inferences.

At the end of the survey, participants were told that they could unfollow the accounts and were asked how likely they were to unfollow them. Fifty-one per cent of participants in the treatments reported being 'very unlikely' or 'unlikely' to unfollow the accounts. Qualitative analyses of participants' comments to the open-ended box at the end of the survey also suggest that many participants enjoyed following the news account and will continue to follow them in the future (Supplementary Information).

### Compliance

Independent coders rated all the screenshots uploaded by the participants in wave 1. We found that compliance for the account following

was high: 87.9% of the participants uploaded screenshots showing they follow both accounts and 92.8% of participants uploaded at least a screenshot showing that they follow one account. Compliance for the notifications was lower: 48.4% of participants uploaded screenshots showing that they activated the notifications for both accounts and 65.1% of participants uploaded at least a screenshot showing that they activated the notifications for one account.

At the beginning of wave 2, participants had to write down the names of the accounts they were asked to follow in wave 1 without looking them up: 71% of participants in the treatments correctly mentioned both news accounts. Then, participants were presented with a list of seven social media accounts and had to click on the ones that they follow on Instagram and WhatsApp. While in wave 1 we used these questions as a screener, in wave 2 we used them as a compliance check. Eighty-nine per cent of participants in the treatments reported following the news accounts (compared with 0% in wave 1). Participants were explicitly asked whether they stopped following the accounts (knowing that their response would be anonymous and would not affect their payment), 13% of participants in the treatments reported having done so. In Supplementary Information we report additional measures of compliance showing that overall compliance was high.

### Statistical analyses

We ran linear mixed-effect models with participants as a random effect. We preregistered adding wave as a random slope (Wave|ID) but the models failed to converge. We analysed the responses at the level of the wave participant, that is, two data points per participant, one per wave. We included an interaction term between 'Condition' and 'Wave' to estimate treatment effects. This approach allows us to determine whether, for example, within-individual gains in current affairs knowledge between waves are larger in the treatments than in the controls. In all analyses, we controlled for age, gender, education and past vote. All statistical tests are two tailed.

Following the preregistration, the main analyses are conducted across countries and platforms, but in Fig. 3 and Supplementary Information we report all treatment effects per country platforms.

### Reporting summary

Further information on research design is available in the Nature Portfolio Reporting Summary linked to this article.

## Data availability

The preregistrations, materials, R scripts and the data to replicate the findings are publicly available via OSF at https://osf.io/8tzd2/ (ref. 79).

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

## Acknowledgements

The author(s) gratefully acknowledge the support of the European Research Council (ERC), 'Europeans exposed to dissimilar views in the media: investigating backfire effects,' Proposal EXPO-756301 (ERC Starting Grant, M.W., PI) and of the Center for Excellence in Social Sciences at the University of Warsaw, which funded the project. The postdoctoral positions of S.A. and E.H. are funded by the ERC under the European Union's Horizon 2020 research and innovation programme (grant agreement no. 883121; PI F. Gilardi). The funders had no role in study design, data collection and analysis, decision to publish or preparation of the manuscript. The authors are grateful to D. Della Casa and F. Melliger for their research assistance, especially for coding the screenshots. Any opinions, findings, conclusions, or recommendations expressed in this material are those of the author(s) and do not necessarily reflect the views of the ERC.

## Author contributions

Original idea by E.H. Conceptualization by S.A., E.H. and M.W. Analyses by S.A. Investigation by S.A. and E.H. Visualization by S.A. Writing—original draft by M.W. Writing—review, editing and revisions by S.A., E.H. and M.W. A coin flip determined that S.A., and not E.H., is corresponding author in addition to M.W. The main purpose of this move is to make the article open access.

## Funding

## Competing interests

M.W. is part of the US 2020 Facebook and Instagram Election Study, which is a pro bono and fully independent research collaboration of researchers with Meta, and was the PI on two unrelated, unrestricted research grants (US$100k and US$50k) from Meta in 2019 and 2021 (the second with E.H. as the co-PI). Meta did not contribute in any way to this study. The remaining authors declare no competing interests.

## Additional information

**Correspondence and requests for materials** should be addressed to Sacha Altay or Magdalena Wojcieszak.

# Reporting Summary

## Statistics

For all statistical analyses, confirm that the following items are present in the figure legend, table legend, main text, or Methods section.

| n/a | Confirmed | |
|---|---|---|
| ☐ | ☒ | The exact sample size (*n*) for each experimental group/condition, given as a discrete number and unit of measurement |
| ☐ | ☒ | A statement on whether measurements were taken from distinct samples or whether the same sample was measured repeatedly |
| ☐ | ☒ | The statistical test(s) used AND whether they are one- or two-sided<br>*Only common tests should be described solely by name; describe more complex techniques in the Methods section.* |
| ☐ | ☒ | A description of all covariates tested |
| ☐ | ☒ | A description of any assumptions or corrections, such as tests of normality and adjustment for multiple comparisons |
| ☐ | ☒ | A full description of the statistical parameters including central tendency (e.g. means) or other basic estimates (e.g. regression coefficient) AND variation (e.g. standard deviation) or associated estimates of uncertainty (e.g. confidence intervals) |
| ☐ | ☒ | For null hypothesis testing, the test statistic (e.g. *F*, *t*, *r*) with confidence intervals, effect sizes, degrees of freedom and *P* value noted<br>*Give P values as exact values whenever suitable.* |
| ☒ | ☐ | For Bayesian analysis, information on the choice of priors and Markov chain Monte Carlo settings |
| ☒ | ☐ | For hierarchical and complex designs, identification of the appropriate level for tests and full reporting of outcomes |
| ☐ | ☒ | Estimates of effect sizes (e.g. Cohen's *d*, Pearson's *r*), indicating how they were calculated |

*Our web collection on statistics for biologists contains articles on many of the points above.*

## Software and code

Policy information about availability of computer code

| Data collection | We used Qualtrics to implement the survey |
|---|---|
| Data analysis | We used R and R studio to conduct the statistical analyses |

For manuscripts utilizing custom algorithms or software that are central to the research but not yet described in published literature, software must be made available to editors and reviewers. We strongly encourage code deposition in a community repository (e.g. GitHub). See the Nature Portfolio guidelines for submitting code & software for further information.

## Data

Policy information about availability of data

All manuscripts must include a data availability statement. This statement should provide the following information, where applicable:
- Accession codes, unique identifiers, or web links for publicly available datasets
- A description of any restrictions on data availability
- For clinical datasets or third party data, please ensure that the statement adheres to our policy

The pre-registrations, materials, R scripts, and the data to replicate the findings are publicly available on OSF at: https://osf.io/8tzd2/

# Research involving human participants, their data, or biological material

Policy information about studies with **human participants or human data**. See also policy information about **sex, gender (identity/presentation), and sexual orientation** and **race, ethnicity and racism**.

| | |
|---|---|
| Reporting on sex and gender | We only asked Gender in the survey and report no gender analysis in the manuscript |
| Reporting on race, ethnicity, or other socially relevant groupings | we did not measure race/ethnicity |
| Population characteristics | In 2024, between March 1 and March 11, we recruited 2,009 German and 2,021 French participants via the mar- ket research company Bilendi. Participants were recontacted between March 18 and 28, 1,700 German participants (985 women, mean age = 40.5 (12.5), median education = A two- year college degree, 38% had a bachelor degree or more), and 1,695 French participants completed the second wave (1,042 women, mean age = 44.1 (11.5), median education = finished high school, 31% had a bachelor degree or more). The median distance between waves was 14 days (M = 13.4, SD = 1.74). InAppendix I we report all exclusions. |
| Recruitment | They were recruited via a Survey company called Bilendi.<br><br>Appendix I.1. France<br>In France, in Wave 1, a total of 8051 participants took the survey.<br>2097 participants did not pass the initial screens and were excluded at the very beginning of the survey because they did not report having a WhatsApp or an Instagram account (715), reported never using WhatsApp and Instagram (138), or were already following one of the social media accounts (1244).<br>2024 participants passed the screens but voluntarily ended the survey on the second consent form, when being told that they would have to follow two social media accounts on What- sApp/Instagram for two weeks. Most of these participants re- ported not wanting to follow new accounts on social media for two weeks – and only 17% reported that the compensation was too low.<br>1909 participants passed the screens and filled out both con- sent forms but did not finish the survey. Most of them (1731) left the survey at the very end when asked to follow the ac- counts and upload the screenshots. These participants were not re-contacted in Wave 2 as uploading the screenshots was a necessary condition to finish the survey and be eligible for Wave 2. The distribution of these participants across condi- tions is similar to the distribution of participants who finished the first wave. There is no sign of differential attrition across Controls/Treatments.<br>Appendix I.2. Germany<br>In Germany, in Wave 1, a total of 8009 participants took the survey.<br>1420 participants did not pass the initial screens and were excluded at the very beginning of the survey because they did not report having a WhatsApp or an Instagram account (225), reported never using WhatsApp and Instagram (149), or were already following one of the social media accounts (1046).<br>2859 participants passed the screens but voluntarily ended the survey on the second consent form, when being told that they would have to follow two social media accounts on What- sApp/Instagram for two weeks. Most of these participants re- ported not wanting to follow new accounts on social media fortwo weeks – and only 15% reported that the compensation was too low.<br>1721 participants passed the screens and filled out both con- sent forms but did not finish the survey. Most of them (1611) left the survey at the very end when asked to follow the ac- counts and upload the screenshots. These participants were not re-contacted in Wave 2 as uploading the screenshots was a nec- essary condition to finish the survey. The distribution of these participants across Control/Treatment is similar to the distribu- tion of participants who finished the first wave. There is no sign of differential attrition across Controls/Treatments. However, participants in the WhatsApp conditions were more likely to drop out than participants in the Instagram conditions. Suggest- ing that in Germany participants may have struggled to follow the accounts on WhatsApp. Such differential attrition between WhatsApp/Instagram groups is not problematic for causal in- ference given that it does not impede the Controls/Treatments randomization – i.e., the WhatsApp Treatments are compared to the WhatsApp Controls. |
| Ethics oversight | This research project complied with all ethical regulations for research involving human subjects and received ethical ap- proval from the University of Zu̇rich PhF Ethics Committee (ethics approval nr. 23.10.14). All study participants submit- ted informed consent before any data were collected. |

Note that full information on the approval of the study protocol must also be provided in the manuscript.

# Field-specific reporting

Please select the one below that is the best fit for your research. If you are not sure, read the appropriate sections before making your selection.

☐ Life sciences ☒ Behavioural & social sciences ☐ Ecological, evolutionary & environmental sciences

For a reference copy of the document with all sections, see nature.com/documents/nr-reporting-summary-flat.pdf

# Behavioural & social sciences study design

All studies must disclose on these points even when the disclosure is negative.

| | |
|---|---|
| Study description | Quantitative. It's a two wave survey, and in between waves we implemented a field experiment by requiring participants to follow two accounts on social media |
| Research sample | In 2024, between March 1 and March 11, we recruited 2,009 German and 2,021 French participants via the mar- ket research company Bilendi. Participants were recontacted between March 18 and 28, 1,700 German participants (985 women, mean age = 40.5 (12.5), median education = A two- year college degree, 38% had a bachelor degree or more), and 1,695 French participants completed the second wave (1,042 women, mean age = 44.1 (11.5), median education = finished high school, 31% had a bachelor degree or more). The median distance between waves was 14 days (M = 13.4, SD = 1.74). InAppendix I we report all exclusions. |
| Sampling strategy | Random. Sample size was determined by budget mostly |
| Data collection | Online, via Qualtrics |
| Timing | see above |
| Data exclusions | see above |
| Non-participation | see above |
| Randomization | Participants first consented to take part in the study and an- swered screening questions about (i) whether they have Insta- gram and WhatsApp accounts, (ii) how frequently they use them, and (iii) whether they follow a list of social media ac- counts. To be eligible, participants had to have either an In- stagram account or WhatsApp, use Instagram or WhatsApp, and not already be following the news accounts included in the experiment. After passing the screens, Instagram users were randomly assigned to the Instagram Control or the Instagram Treatment, WhatsApp users were randomly assigned to the WhatsApp Control or the WhatsApp Treatment, and those who used both Instagram and WhatsApp were randomly assigned to one of the four conditions (see Figure 1). |

# Reporting for specific materials, systems and methods

We require information from authors about some types of materials, experimental systems and methods used in many studies. Here, indicate whether each material, system or method listed is relevant to your study. If you are not sure if a list item applies to your research, read the appropriate section before selecting a response.

## Materials & experimental systems

| n/a | Involved in the study |
|---|---|
| ☒ | ☐ Antibodies |
| ☒ | ☐ Eukaryotic cell lines |
| ☒ | ☐ Palaeontology and archaeology |
| ☒ | ☐ Animals and other organisms |
| ☒ | ☐ Clinical data |
| ☒ | ☐ Dual use research of concern |
| ☒ | ☐ Plants |

## Methods

| n/a | Involved in the study |
|---|---|
| ☒ | ☐ ChIP-seq |
| ☒ | ☐ Flow cytometry |
| ☒ | ☐ MRI-based neuroimaging |

# Plants

| | |
|---|---|
| Seed stocks | *Report on the source of all seed stocks or other plant material used. If applicable, state the seed stock centre and catalogue number. If plant specimens were collected from the field, describe the collection location, date and sampling procedures.* |
| Novel plant genotypes | *Describe the methods by which all novel plant genotypes were produced. This includes those generated by transgenic approaches, gene editing, chemical/radiation-based mutagenesis and hybridization. For transgenic lines, describe the transformation method, the number of independent lines analyzed and the generation upon which experiments were performed. For gene-edited lines, describe the editor used, the endogenous sequence targeted for editing, the targeting guide RNA sequence (if applicable) and how the editor was applied.* |
| Authentication | *Describe any authentication procedures for each seed stock used or novel genotype generated. Describe any experiments used to assess the effect of a mutation and, where applicable, how potential secondary effects (e.g. second site T-DNA insertions, mosiacism, off-target gene editing) were examined.* |

