## [Peer Review file · Nature Human Behaviour]

Following News on Social Media Boosts Knowledge, Belief Accuracy, and Trust

Corresponding Author: Professor Magdalena Wojscieszak

Version 0:

Decision Letter:

9th October 2024

Dear Dr Altay,

Thank you once again for your manuscript, entitled "News on Social Media Boosts Knowledge, Belief Accuracy, and Trust: A Field Experiment on Instagram and WhatsApp," and for your patience during the peer review process.

Your manuscript has now been evaluated by 3 reviewers, whose comments are included at the end of this letter. Although the reviewers find your work to be of interest, they also raise some important concerns. We are interested in the possibility of publishing your study in Nature Human Behaviour, but would like to consider your response to these concerns in the form of a revised manuscript before we make a decision on publication.

In particular, we believe it will be important to resolve Reviewer #2's concerns about the ethics of this study (point 8).

In sum, we invite you to revise your manuscript taking into account all reviewer and editor comments. We are committed to providing a fair and constructive peer-review process. Do not hesitate to contact us if there are specific requests from the reviewers that you believe are technically impossible or unlikely to yield a meaningful outcome.

We hope to receive your revised manuscript within two months. I would be grateful if you could contact us as soon as possible if you foresee difficulties with meeting this target resubmission date.

- Include a "Response to the editors and reviewers" document detailing, point-by-point, how you addressed each editor and referee comment. If no action was taken to address a point, you must provide a compelling argument. When formatting this document, please respond to each reviewer comment individually, including the full text of the reviewer comment verbatim followed by your response to the individual point. This response will be used by the editors to evaluate your revision and sent back to the reviewers along with the revised manuscript.
- Highlight all changes made to your manuscript or provide us with a version that tracks changes.

Link Redacted

We look forward to seeing the revised manuscript and thank you for the opportunity to review your work. Please do not hesitate to contact me if you have any questions or would like to discuss these revisions further.

Sincerely,

██████████
██████████
██████████
Nature Human Behaviour

REVIEWER COMMENTS:

Reviewer #1 (Remarks to the Author):

This manuscript presents a nicely done field experiment on the effect of (potential) exposure to news on Instagram and WhatsApp over a two-week period. The authors find a positive effect on knowledge, news discernment and awareness, and news trust. I think the study has high internal validity - compliance with the treatment was high, and the placebo design likely helped guard against differential attrition. As such, I don't have major issues and I would be happy to see this published in NHB.

Two interesting aspects that would be worth developing further: (1) I agree that notifications may be an important part of the story here - what are the implications for our understanding of incidental exposure to news? Is this more about awareness of news than about full understanding of events behind the news (which may require more sustained reading/viewing of stories)? (2) Should our theoretical expectations differ for effects on IG vs. WhatsApp? For example, the latter notably does not use an algorithm to rank content, which may have consequences for likelihood of exposure.

A few smaller points below:

- What was the exact wording used for the trust questions? The effect size here is large, which may or may not be surprising depending on (a) how crystallized participants' attitudes toward the news were (b) whether the q's were geared toward the sources in general or on their platform-specific manifestations, about which attitudes may have been less well formed.
- The authors argue that there is "a demand for free and reliable news on social media platforms". The evidence they present is suggestive, though perhaps in tension with their null findings on interest in news and politics.
- The idea that "users were largely unaware that it is easy and free to follow quality news media organizations on social media platforms" seems related to the findings of Chen and Yang (2019), who show that prior to their experiment, participants were largely unfamiliar with the extent of information that was available via VPNs.
- Was the experiment conducted before IG fully rolled out its algorithm change to turn off recommendations to non-followed political accounts by default? ("In the Instagram groups, 29% of participants reported never consuming news on Instagram and 23% reported consuming it less than once a day," perhaps suggesting relatively high levels?)
- Typo Figure 2 caption: "orange squares are the estimates of the German WhatsApp treatment" -> orange triangles

Reviewer #2 (Remarks to the Author):

This is an interesting and potentially exciting paper. I applaud the authors for this undertaking. Below, I articulate several comments and concerns that I would like to see addressed before considering this paper for publication in Nature Human Behaviour.

1. I was somewhat convinced when the authors stated how they have extended the literature in three important ways. These are practical or functional extensions/rationales of how they have added to science. The third rationale is somewhat concerning in the sense that these social media platforms are not typically used for news (as the authors indicate). Therefore, if we forget about the specific platforms and focus instead on their features or affordances, this might be more compelling. What features or affordances define each platform, and then what do we know about platforms of this type within the context of your results? In short, I'm interested in seeing if the authors can take their thinking one level up in abstraction to make broader conclusions about social media platforms, in general, not just these two specific ones that were under investigation. Instagram and WhatsApp are wildly different, so focusing on features and affordances are likely important for generalizations (if applicable). This would help to position the work in media psychology research as well.

2. I would also like to see the authors go beyond practical and function rationales. Why is this work interesting, informative, and impactful from a theory/theoretical perspective? Evidence from social psychology and behavioral economics finds differential effects of framing depending on whether people focus on good (gains) vs. bad (losses), for example. Is this an appropriate way to position the research, theoretically? For the detection accuracy part, there's an entire literature on deception detection (see Duped and truth-default theory by Levine) that can speak to this research. For real vs. fake news detection, research by Luo et al (2022) in Communication Research may provide grounding as well. Altogether, the authors need to couch this work in theories and theoretical ideas that are bigger, more encompassing than the phenomenon itself.

Right now, the authors are all about the effect(s). While this is common for general science outlets, there should be some clear theoretical application and extension here as well.

3. Some clarifications: how did the authors ensure “participants had to use WhatsApp or Instagram, and not already be following the news accounts included in the experiment.” Any validity checks? Was the screenshot approach applied here as well?

4. I may have missed this, but I have a question regarding how often news vs. non-news posts occurred and were seen by participants. How often did people actually see content from the news or non-news sites? With uncertainty regarding the algorithms that underly these platforms, the induction is also therefore uncertain. Did people see news and non-news posts once per day? Five times per day? Was this posting frequency equivalent? Results will most certainly be confounded with the base-rate of posting for each platform. Perhaps, news accounts were posting more (or less) than non-news accounts, impacting these reported effects. In other words, are these by-condition results attributed to a news effect, or attributed to a frequency of how often information is presented to a person effect? Much more information about what people actually saw is required. This might be remedied, and easily analyzable, by controlling for account posting frequency with the data collection time period as a fixed effect in your linear mixed models.

5. The finding that “the treatments increased gains in awareness of true news stories (but not false news stories), namely the number of news stories participants reported having heard of or read before – while controlling for participants’ tendency to report being aware of news stories they have not encountered with a placebo news story” would be informed by deception theory, namely truth-default theory. Along these lines, what was overall detection accuracy and the rate of the truth-bias (the proportion of truthful judgments divided by the total number of judgments)? Unpacking the detection accuracy results more will be helpful.

6. A cynical reader (not myself, for the record) would say “so what?” to these findings. That is, if you ask people to follow (good) news compared to non-news, no wonder they had positive effects on news-related outcomes! The longitudinal, multi-site, multi-platform design — that occurred in the wild — counterbalances this cynical view. Therefore, I’d like to see the authors articulate the impact and value of this work more in the Discussion. Tell the readers why this work is practically and theoretically compelling beyond the obvious methodological improvements on prior work.

7. One of the recommendations offered or implied by the authors is that following good news may help people achieve democratic ideals. Who defines what is “good” here? I am writing from the US, and we are quite divided in America over the media. To those who are left-leaning and liberal, the NYTimes and MSNBC are often perceived as “good,” and Fox News, Breitbart, etc. are “bad.” To those who are right-leaning and conservative, the opposite is true. Goodness or badness is therefore (unfortunately) subjective, and not objective. There are media bias scores that might indicate goodness or badness (<https://adfontesmedia.com/interactive-media-bias-chart/>), but these perceptions are in the eye of the news consumer. How do the authors contend with the idea that you simply will not get a very right-leaning and conservative person on social media to follow the NYTimes? See recent Pew data: <https://www.pewresearch.org/journalism/2014/10/21/political-polarization-media-habits/>. Of course, the experimental results in this work had random assignment so any ideology effect is likely accounted for in the design. But, practically speaking, can this study — plus the ethos and design behind it — actually inform or intervene in the ways the authors idealize? This is curious to me, and I am skeptical. Perhaps this would work in other countries that are less divided than the US.

8. Can the authors elaborate on the ethics approval of this study and the ethical considerations of presenting people with fake/false news? Yes, this is a research study and presumably participants may have known that they would be presented with some true or untrue content. But, perhaps people believed what they saw. This presents a major ethical issue if proper debriefing and discussion of the experimental materials were not performed (e.g., people communicating to their friends, after the study, about what a global crisis). Please elaborate and write about this at length in the manuscript.

Altogether, I find the work interesting, but I need more from the authors. I look forward to seeing them take up the gauntlet in a possible revision.

Reviewer #3 (Remarks to the Author):

This paper addresses the ongoing debate on whether people can learn from or via social media about political events and actors. Although, the paper is not novel in terms of concept or theory, it uses a very innovative approach in terms of data and methods. In contrast to most studies on this topic it uses a field experiment to expose individuals to news on different social media platforms. By incentivizing social media users in France and Germany to follow quality/mainstream news outlets on Instagram and WhatsApp for two weeks. In this way the study does not try to correct the ‘bad’ information out there, but focuses on ‘promoting’ reliable information about recent political events. The use of compliance measures in the study is innovative and really well done.

Overall, the data approach seems valid to me, gathering high quality data, and also the presentation of the empirical findings is clear and solid. I believe the authors are a bit too ‘positive’ in their interpretation of the data. Not all results are significant (depending on platform and country), but more importantly the effects seems rather small (taking into account the relative large N). A bit more attention to the effect sizes would be insightful.

In the method section (4.9) the authors mention they control for age, gender, education, and past vote. But what about

controlling for media use? I guess the effects are most outspoken for those that have a low news diet? (and perhaps a ceiling effect for news junkies?). This seems more important to me than controlling for past vote. (News use is mentioned as a moderator in the preregistration of W1)

The Preregistration is extensive and clear. All the deviations from the pre-registered protocol are well documented and make sense.

In the conclusion the authors make a convincing claim that incentivizing social media users to follow news on Instagram and WhatsApp can lead to knowledge gains, and other positive democratic outcomes. However, without disputing these 'hopeful' results, the authors might be a bit more critical. The paper stresses the possibility of a virtuous circle between news consumption on social media an algorithmic suggestions for more news, but nothing is mentioned about a potentially more vicious circle where people decide to follow more alternative/partisan or low quality news outlets. And what if people decide to no longer subscribe to a newspaper and prefer the short/free news updates on whatsApp? Again, I think the potential positive effects are real, but a more nuanced discussion of these effects seems warranted.

Next, I believe the discussion could be more relevant for scholars working on this topic if the authors would interact more with previous studies. Some studies (cited in this study) have speculated about the non-findings of social media on political knowledge and suggested the lack of actual news content on social media as one of the potential explanations (eg. Boukes; Van Erkel & Van Aelst). This study provides actual prove that this is one of the main drivers. People perceive that they have seen news stories on social media, but in reality they often have not (or not enough).

Additional points for improvement:

WhatsApp is treated as a social medium in this paper. Most studies label it as a (private) messenger app. Please clarify why WhatsApp is considered a social media platform.

I find the term 'news knowledge' a bit confusing. In my view the authors measure political knowledge, or what in the literature is labelled as 'surveillance political knowledge' (or current affaires knowledge). It is about recent political events and actors that were featured in news coverage. The authors also sometimes speak about political knowledge, so it would be more consistent to use this term (note: in the preregistration the term surveillance knowledge is used). In that regard, it might be useful to document the knowledge questions a bit more in the appendix. What was the number of correct answers in wave 1 and wave2? To what extent were the answers to this questions mentioned in the news outlets used in the experiment (in the two weeks period)? In other words, how much opportunities were there to learn? It might be that the question about the current Minister of National Education and Youth? (W1 and W2 FRANCE) was not given news attention, so you would not expect people to improve their knowledge (and in that case the term 'news knowledge' is simply incorrect).

Overall, this is a truly innovative study that is well executed and provides relevant results. My suggestions for improvement are rather minor.

Version 1:

Decision Letter:

Our ref: NATHUMBEHAV-24072962A

4th February 2025

Dear Dr. Wojscieszak,

Thank you for submitting your revised manuscript "News on Social Media Boosts Knowledge, Belief Accuracy, and Trust: A Field Experiment on Instagram and WhatsApp" (NATHUMBEHAV-24072962A). It has now been seen by the original referees and their comments are below. As you can see, the reviewers find that the paper has improved in revision. We will therefore be happy in principle to publish it in Nature Human Behaviour, pending minor revisions to satisfy the referees' final requests and to comply with our editorial and formatting guidelines.

We are now performing detailed checks on your paper and will send you a checklist detailing our editorial and formatting requirements within two weeks. Please do not upload the final materials and make any revisions until you receive this additional information from us.

Sincerely,

██████████

██████████████████

██████████████████

Nature Human Behaviour

Reviewer #1 (Remarks to the Author):

The authors have done a great job addressing my comments. I'm happy with the revised manuscript and would look forward

to seeing it published.

Reviewer #2 (Remarks to the Author):

I applaud the authors for their attention to the requests in this review. I have surveyed all of their responses across reviewers, and I am pleased that they have addressed all points. I now believe this paper is ready for publication. However, for historical accuracy, I also believe they need to edit one line on the bottom p. 6 where they state "Israel's attacks on Gaza." This should be "Israel's attacks on Gaza following the October 7th attacks by Hamas."

Reviewer #3 (Remarks to the Author):

I believe the authors did a good job addressing all the comments and concerns. My smaller suggestions are all addressed, so I have no new fundamental comments and look forward to the publication of this important study.

I have one minor point on the framing of the outcomes in the abstract. In particular the following sentences: "These results suggest that, contrary to popular belief, people actually learn from news on social media and that this form of news consumption should not be dismissed or disdained. While some forms of social media use are harmful, others are positive and can be leveraged to foster a well-informed society." Two (related) things are slightly misleading to me. First the notion of 'popular belief' suggests that this notion of limited learning is not based on previous research. Second, the abstract does not mention that people learn from social media if they are presented with more actual news content as in this natural experiment. Therefore, I think the last sentence of the discussion section is more correct. "This means that, with the right incentives social media can be a powerful tool for promoting informed and engaged citizenship."

Reviewer 1

Reviewer 1: This manuscript presents a nicely done field experiment on the effect of (potential) exposure to news on Instagram and WhatsApp over a two-week period. The authors find a positive effect on knowledge, news discernment and awareness, and news trust. I think the study has high internal validity - compliance with the treatment was high, and the placebo design likely helped guard against differential attrition. As such, I don't have major issues and I would be happy to see this published in NHB.

Thank you for noting that our manuscript presents a nicely done field experiment and that the study has high internal validity. Thank you also for your useful comments and queries. We address each of these issues below.

Reviewer 1: Two interesting aspects that would be worth developing further: (1) I agree that notifications may be an important part of the story here - what are the implications for our understanding of incidental exposure to news? Is this more about awareness of news than about full understanding of events behind the news (which may require more sustained reading/viewing of stories)? (2) Should our theoretical expectations differ for effects on IG vs. WhatsApp? For example, the latter notably does not use an algorithm to rank content, which may have consequences for likelihood of exposure.

Authors: Regarding (1), you are right that we do not measure deep understanding of news events but rather 'political surveillance knowledge', i.e., relatively superficial knowledge about current affairs. We are now clearer about this in the manuscript and use the term 'current affairs knowledge'.

"Our current affairs knowledge questions do not measure how well participants understand current political events, but rather their knowledge related to current affairs." p. 8

Notifications are very likely to facilitate incidental new exposure, as with notifications people do not need to open an app to be incidentally exposed to news, and simply have to look at the locked screen of their smartphone. This, however, assumes that notifications contain news content or current affairs information. While it is the case for the WhatsApp notifications, it is not true for Instagram notifications – as they simply inform people about the activity of an account rather than the content of that activity. The notifications may have enhanced treatment effects simply because they acted as a news reminder and facilitated access to news content. We now discuss this briefly in the manuscript. Note, however, that because we don't detect meaningful differences between the WhatsApp and Instagram treatments, we do not speculate too much on the role that these differences in notifications may have played or not.

"Future work should also investigate the theoretical mechanisms behind the treatment effects. Breaking down the results by compliance levels offers some preliminary insights. The absence of statistically significant effects among participants who followed the news accounts but did not turn on the notifications (compared to participants who turned them on; the full compliers) suggests that notifications may have played an important role. Notifications could expose users to news content directly, cutting through the in-platform noise and informing users about

current events, and/or serve as a mere reminder for the users to access news. News notifications are not unusual. In Wave 1, 47.5% of participants reported having received news notifications on their phones in the past week (in line with \citep{lu2016more}).” p.6

Regarding (2), we now discuss how theoretical expectations may differ or not between WhatsApp and Instagram, based on the affordances of these platforms.

“Although this project offers new optimistic evidence, we encourage future research to examine the generalizability of our findings. For one, it is unclear whether similar effects would emerge on platforms with different features or uses. In our case, we do not detect statistically significant differences between platforms, but we lack statistical power to detect small differences. Yet, there are reasons to expect that platforms with different algorithms and affordances would yield distinct results. For instance, if participants had followed news accounts on TikTok, they may never have been exposed to content from these accounts because users mostly use the "For you" feed which weakly prioritizes followed accounts. Similarly on YouTube, where the algorithms direct users away from news (Huang and Yang, 2024), subscribing to news channels may not increase news exposure in any pronounced ways (but tweaking the algorithm itself does generate these increases (Yu et al., 2023a). However, holding the quantity and quality of news constant, exposure to news on TikTok, YouTube or other platforms should also yield the benefits documented here.” p.6

Reviewer 1: A few smaller points below:

- What was the exact wording used for the trust questions? The effect size here is large, which may or may not be surprising depending on (a) how crystallized participants' attitudes toward the news were (b) whether the q's were geared toward the sources in general or on their platform-specific manifestations, about which attitudes may have been less well formed.

Authors: Thank you for this good question! The exact wording is “To what extent do you trust the media and journalists?” (translated from the French “Dans quelle mesure faites-vous confiance aux médias et journalistes ?” and German “Wie gross ist Ihr Vertrauen in die Medien und Journalisten?”). We now report it in the method:

“[...] the extent to which they trust the news and journalists (“To what extent do you trust the media and journalists?” from 'Not at all' [1] to 'Completely' [7]) [...].”

In Appendix F, we now provide descriptive statistics for each dependent variable (p. 16).

The trust question is not platform or source specific – although note that we find similar effect sizes for this general trust question and for the specific trust questions about the news outlets that participants were asked to follow (e.g., “To what extent do you trust France Info?”).

It is hard to know what exactly is driving this effect, but we have added a potential explanation in the discussion:

“[...] These effects are (likely) due to increased exposure to news content: participants in the treatment saw more news and learned from it. The effects on trust may be due to

participants underestimating the quality of news coverage before the experiment and updating their perceptions after being exposed to this coverage---which in turn--- spilled over to perceptions of news and journalists more broadly.”

Reviewer 1: - The authors argue that there is "a demand for free and reliable news on social media platforms". The evidence they present is suggestive, though perhaps in tension with their null findings on interest in news and politics.

Authors: We have added a sentence to better back-up the demand for news on social media:

“Such positive incentivizations align with users’ desire for accurate information and educational content, instead of divisive, hateful, or false content (Rathje et al., 2023)” p.2

We also note that people find it convenient to consume news on social media, and that people around the world increasingly consume news on social media:

“People increasingly consume news on social media, and although they worry about inaccurate news, they find it convenient to use social media for news (Newman et al., 2024; Wang and Forman-Katz, 2024).” p.6

It is not clear to us why the fact that the treatment did not increase news/political interest is incompatible with the fact that there is an unmet demand for news. It is possible that people would like to see slightly more quality news on social media, but that being exposed to quality news on social media does not increase interest in the news. If exposure to news increased interest it may suggest that exposure created demand and that demand did not precede exposure.

Reviewer 1: - The idea that "users were largely unaware that it is easy and free to follow quality news media organizations on social media platforms" seems related to the findings of Chen and Yang (2019), who show that prior to their experiment, participants were largely unfamiliar with the extent of information that was available via VPNs.

Authors: Thank you for directing us to this piece, which we now include in the final article.

“These qualitative insights suggest that the lack of awareness about the presence of quality news on platforms may partly contribute to the low levels of news consumption on social media (similar to the unawareness of censored information available via VPNs in China (Chen and Yang, 2019)).” p.5

Reviewer 1:- Was the experiment conducted before IG fully rolled out its algorithm change to turn off recommendations to non-followed political accounts by default? ("In the Instagram groups, 29% of participants reported never consuming news on Instagram and 23% reported consuming it less than once a day," perhaps suggesting relatively high levels?)

Authors: The experiment was conducted between the March 1st 2024 and March 28th 2024, which was after Meta’s decision to not recommend content about politics on Instagram, which was announced February 9th 2024: “If you decide to follow accounts that post political content, we don’t want to get between you and their posts, but we also don’t want to proactively recommend political content from accounts you don’t follow. [...] we won’t proactively recommend

content about politics on recommendation surfaces across Instagram and Threads. If you still want these posts recommended to you, you will have a control to see them.”

<https://about.instagram.com/blog/announcements/continuing-our-approach-to-political-content-on-instagram-and-threads>

Given the lack of transparency, it is not clear to us whether --- and when --- this algorithmic change was actually deployed in France and Germany. In any case, our results are certainly contextual to the design of the platforms as well as their algorithms at the moment of the experiment. If Meta was not downgrading news and political information on its platforms at the time of the experiment, our treatment effects would probably have been weaker, as news and political information would have been more prevalent at baseline. We are now clearer about this and mention it in the discussion:

“Moreover, our findings are contextual to the design of the platforms and their algorithms at the moment of the experiment (Munger, 2019), and so extending our work to different platforms, countries, and time periods is needed” p.6

Reviewer 1:- Typo Figure 2 caption: "orange squares are the estimates of the German WhatsApp treatment" -> orange triangles

Authors: Thank you for spotting this typo, we fixed it.

In general, we greatly appreciate all the positive words and the useful suggestions. We hope that these changes and additions fully address all of your comments.

Reviewer 2

Reviewer 2: This is an interesting and potentially exciting paper. I applaud the authors for this undertaking. Below, I articulate several comments and concerns that I would like to see addressed before considering this paper for publication in Nature Human Behaviour.

Thank you for noting this manuscript is interesting and potentially exciting. We also appreciate the many suggestions which we have addressed.

Reviewer 2: 1. I was somewhat convinced when the authors stated how they have extended the literature in three important ways. These are practical or functional extensions/rationales of how they have added to science. The third rationale is somewhat concerning in the sense that these social media platforms are not typically used for news (as the authors indicate). Therefore, if we forget about the specific platforms and focus instead on their features or affordances, this might be more compelling. What features or affordances define each platform, and then what do we know about platforms of this type within the context of your results? In short, I'm interested in seeing if the authors can take their thinking one level up in abstraction to make broader conclusions about social media platforms, in general, not just these two specific ones that were under investigation. Instagram and WhatsApp are wildly different, so focusing on features and affordances are likely important for generalizations (if applicable). This would help to position the work in media psychology research as well.

Authors: Thank you for this comment, which we address in two ways. First, we substantially rewrote the front end and the third justification as to how we expand past work to account for differential uses and affordances. Second, we expand the discussion section to further speculate

how the findings may differ across platform designs. Please note that due to space limitations, we are not able to engage with this in great detail; nevertheless we hope that the additions and clarifications are sufficient and satisfactory.

“Although this project offers new optimistic evidence, we encourage future research to examine the generalizability of our findings. For one, it is unclear whether similar effects would emerge on platforms with different features or uses. In our case, we do not detect statistically significant differences between platforms, but we lack statistical power to detect small differences. Yet, there are reasons to expect that platforms with different algorithms and affordances would yield distinct results. For instance, if participants had followed news accounts on TikTok, they may never have been exposed to content from these accounts because users mostly use the "For you" feed which weakly prioritizes followed accounts. Similarly on YouTube, where the algorithms direct users away from news (Huang and Yang, 2024), subscribing to news channels may not increase news exposure in any pronounced ways (but tweaking the algorithm itself does generate these increases (Yu et al., 2023a). However, holding the quantity and quality of news constant, exposure to news on TikTok, YouTube or other platforms should also yield the benefits documented here.”
p.6

Reviewer 2: 2. I would also like to see the authors go beyond practical and function rationales. Why is this work interesting, informative, and impactful from a theory/theoretical perspective? Evidence from social psychology and behavioral economics finds differential effects of framing depending on whether people focus on good (gains) vs. bad (losses), for example. Is this an appropriate way to position the research, theoretically? For the detection accuracy part, there's an entire literature on deception detection (see Duped and truth-default theory by Levine) that can speak to this research. For real vs. fake news detection, research by Luo et al (2022) in Communication Research may provide grounding as well. Altogether, the authors need to couch this work in theories and theoretical ideas that are bigger, more encompassing than the phenomenon itself. Right now, the authors are all about the effect(s). While this is common for general science outlets, there should be some clear theoretical application and extension here as well.

Authors: We have extended the theoretical implications of our findings in the discussion:

“These two core findings, together, suggest that improving exposure to verified information from news media could be more effective than reducing exposure to low-quality information from junk news. The increased belief in true news and trust in news and journalists is important given the evidence that the rejection of true news may be a bigger problem than the acceptance of false news. For instance, people are skeptical of true news \cite{pfanderAltay2023meta}, distrust news media in general, including reliable news outlets \cite{newman2023digital}, and trust mainstream news outlets much less than fact-checkers do \cite{pennycook2019fighting}. This suggests that improving exposure to reliable information may be more beneficial \cite{askari2024incentivizing, altay2024tips} than reducing exposure to misinformation, which---after all---represents a small portion of the news people see online (\cite{acerbi2022research}).” p.5

“Future work should also investigate the theoretical mechanisms behind the treatment effects. Breaking down the results by compliance levels offers some preliminary insights.

The absence of statistically significant effects among participants who followed the news accounts but did not turn on the notifications (compared to participants who turned them on; the full compliers) suggests that notifications may have played an important role. Notifications could expose users to news content directly, cutting through the in-platform noise and informing users about current events, and/or serve as a mere reminder for the users to access news. News notifications are not unusual. In Wave 1, 47.5% of participants reported having received news notifications on their phones in the past week (in line with \citep{lu2016more}).

The beneficial effects detected may be explained through cognitive processing theories \citep{cacioppo1986central}: the concise, visually-driven nature of short-form news on platforms like Instagram and WhatsApp may enhance information retention by reducing cognitive overload, while the interactive features of these platforms could foster deeper engagement. Unlike traditional, longer formats, the processing of which requires deeper, more reflective processing (per the Elaboration Likelihood Model \citep{petty2011elaboration}), short-form news may rely more on peripheral cues like visuals or source credibility. This could mean that the cognitive and emotional pathways through which these formats influence knowledge retention, trust, and content discernment differ substantially from those activated by traditional media. We encourage scholars to explore these under-studied mechanisms.” p.6

Reviewer 2: 3. Some clarifications: how did the authors ensure “participants had to use WhatsApp or Instagram, and not already be following the news accounts included in the experiment.” Any validity checks? Was the screenshot approach applied here as well?

Authors: Thank you for this good query! We only used screenshots to make sure that participants actually followed the accounts suggested by us as part of the experimental treatment. For the platform use and account following prior to the experiment, we relied on self-reported questions, which were used as filters in the screening survey.

Regarding WhatsApp and Instagram use, we believe there is no reason to distrust participants' self-reported use. Even if the correlation between actual use and self-reported use is imprecise, we do not need a precise estimate of their use. Instead, we only need to know whether participants use these platforms or not. To the extent that participants had no incentive to lie, as they were not told that using these platforms was needed to participate in the study, we have no reasons to believe that participants misreported platform use. We also know that the vast majority of participants had the apps on their phones given that they uploaded screenshots of it.

Regarding the account following, we believe that the data from the self-reported questions in Wave 2 shows that the question is quite precise. Indeed, while 0% of participants reported following the news accounts in wave 1, this percentage jumped to 89% in wave 2. Similarly, more than 75% of participants were able to write down the name of the news accounts based on memory – suggesting that, to some extent, people know whether they follow specific accounts or not.

In any case, non-compliance or inaccurate responses to these questions would reduce our treatment effects, not inflate them. That is, participants already following the news accounts before the experiments, either in the treatment or control, would reduce treatment effect. In short, if some of these self-reports were inaccurate (and -- again -- there is no reason to suspect they were), we would present conservative estimates of the tested effects. We are already above the

words limit, but if deemed necessary we can add a short version of this response in the discussion.

Reviewer 2: 4. I may have missed this, but I have a question regarding how often news vs. non-news posts occurred and were seen by participants. How often did people actually see content from the news or non-news sites? With uncertainty regarding the algorithms that underly these platforms, the induction is also therefore uncertain. Did people see news and non-news posts once per day? Five times per day? Was this posting frequency equivalent? Results will most certainly be confounded with the base-rate of posting for each platform. Perhaps, news accounts were posting more (or less) than non-news accounts, impacting these reported effects. In other words, are these by-condition results attributed to a news effect, or attributed to a frequency of how often information is presented to a person effect? Much more information about what people actually saw is required. This might be remedied, and easily analyzable, by controlling for account posting frequency with the data collection time period as a fixed effect in your linear mixed models.

Authors: Thank you for this good query! Unfortunately, we have no way of knowing how often participants actually saw posts from news or non-news accounts and how often these news accounts posted versus how often other accounts that the participants followed posted during the time of the experiment. Self-reported data in Wave 2 suggests that people in the treatment were exposed to more news on social media, more news on WhatsApp/Instagram, and received more news notifications.

“In Wave 2, participants in the Treatments were more likely to report having received news notifications on their phones in the past week than participants in the Controls. In wave 1, 46% of participants in the Treatment and 49% of participants in the Control reported having received news notifications on their phones, while in wave 2 this percentage goes up to 70% in the treatment and 51% in the Control. These differences are statistically significant ($b = .20, p < .001$).

Self-reported news consumption on WhatsApp significantly increased between waves among participants in the WhatsApp Treatment compared to participants in the WhatsApp Control ($b = .78, p < .001$). In Wave 1, WhatsApp news consumption was slightly higher in the Control than in the Treatment (by .04), whereas in Wave 2 WhatsApp news consumption is much higher in the Treatment than in the Control (by .77).

Self-reported news consumption on Instagram significantly increased between waves among participants in the Instagram Treatments compared to participants in the Instagram Controls ($b = .44, p < .001$). In Wave 1, Instagram news consumption was slightly higher in the Treatment than in the Control (by .01), whereas in Wave 2 Instagram news consumption was much higher in the Treatment than in the Control (by .45).

In Wave 2, participants in the Treatments were more likely to report that social media is their main source of news (21.9% in Wave 1 and 25.1% in Wave 2) compared to participants in the Control (21.3% in Wave 1 and 20.4% in Wave 2; $b = .04, p < .001$).”

Regarding the posting frequency, we selected accounts that post regularly, at least once a day, and tried to match their posting frequency. However, even if non-news accounts post more often

than news accounts (or vice versa), it would not represent a threat to the internal validity of the field experiment – given that we document no case of differential attrition. Note that we added control accounts to avoid randomization threats, in that participants in the control could end up being different from participants in the treatment (e.g., if some participants in the control were unwilling to follow accounts while all participants in the treatment were willing to follow accounts).

It is not clear to us that it would be a good idea to control for posting frequency. Controlling for posting frequency would estimate treatment effects *while holding exposure constant*. This is a different research question than the one that motivated this paper. Posting frequency is likely a causal path through which the accounts have an effect: e.g., if an account does not post, it has no effect, and the more an account posts the more people are likely to see its posts, etc. If measured properly, controlling for posting frequency would reduce the effect of the treatment (as it would be the effect of the treatment holding exposure constant). If you or the editor deem it necessary, we can run such models, but here are some additional reasons why we don't think it would be particularly useful.

(i) Practically, we would have to estimate posting frequency based on the account current posting frequency, as we did not precisely measure it at the time of the experiment. This would provide imprecise estimates.

(ii) We already run moderation analyses that answer this question. Specifically, we test whether participants who report being exposed to more news in Wave 2, compared to Wave 1, show stronger treatment effects. This is likely a more precise estimate of the effect of news exposure on treatment effects than controlling for posting frequency. Indeed, participants self-reported measures of news exposure on Instagram and WhatsApp are likely a better proxy of exposure to news than the accounts posting frequency.

We hope that the moderation analyses using self-reported news exposure, as well as the explanations related to posting frequency, will address your good question and justify our approach. Again, if you or the editor deem it essential, we will happily add posting frequency to the models.

Here are the posting frequency of the accounts, roughly estimated based on their current posting frequency:

“FRANCE, WhatsApp

HugoDecrypte: once a day

FranceInfo: 5-6 posts a day, but mostly one main news post per day, max 3.

Marmiton (control): around 4 posts a day

Allocine (control): around 5-6 posts a day

FRANCE, Instagram

HugoDecrypte: between 1 and 5 posts a day; rarely post stories

FranceInfo: Around 5 posts a day; around 5 stories a day as well.

Inrocks (control): around 2 posts a day; around 4 stories a day.

Musee Rodin (control): less than 1 post per day; 0 stories a day

GERMANY, WhatsApp

SZ: 5-6 posts a day

FAZ: 4-5

Einfach Tasty (control): 1 post a day

Serienjunkies (control): 1 post a day

GERMANY, Instagram

SZ: around 5 posts a day; 5 stories a day

FAZ: around 5 posts a day; barely any stories

Klassik Radio (control): 1 post a day; 2 stories a day

Farbenfroh Art (control): less than 1 post per day; 0 stories a day”

Overall, it seems that accounts in the control post less than accounts in the treatment (except for WhatsApp in France). We believe that it is another reason to not control for posting frequency, as it will be treated as a confounding factor in the models, even though participants cannot possibly learn from the posts of the control, as these posts don't cover the news. We believe it makes little sense to treat posting frequency as a confound (by adding it as a control), however it could be treated as a moderator (by interacting it with treatment). But again, our estimates of posting frequency are noisy and likely less precise than participants' self reported news exposure on the platforms or their self-reported WhatsApp/Instagram use, which we also test as a moderator and has no significant effect.

Reviewer 2: 5. The finding that “the treatments increased gains in awareness of true news stories (but not false news stories), namely the number of news stories participants reported having heard of or read before – while controlling for participants' tendency to report being aware of news stories they have not encountered with a placebo news story” would be informed by deception theory, namely truth-default theory. Along these lines, what was overall detection accuracy and the rate of the truth-bias (the proportion of truthful judgments divided by the total number of judgments)? Unpacking the detection accuracy results more will be helpful.

Authors: Thank you for encouraging us to unpack these interesting results! We have added information about the detection accuracy and the rate of truth-bias in Appendix:

“When looking at additive discernment ($\text{meanTrue} - \text{meanFalse}$):

In Wave 2, in the control, true claims were rated 1.52pts higher than false claims.

In Wave 2, in the treatment, true claims were rated 1.58pts higher than false claims.

When looking at multiplicative discernment ($\text{meanFalse}/\text{meanTrue}$):

In Wave 2, in the control, true claims were rated 1.43 times higher than false claims.

In Wave 2, in the treatment, true claims were rated 1.42 times higher than false claims.

For the measures of discernment we follow: <https://www.nature.com/articles/s41562-023-01667-w>

When looking at truth-bias ($(\text{maxTrue} - \text{meanTrue}) - (\text{meanFalse} - \text{minFalse})$):

In Wave 2, in the control, participants showed a small skepticism bias, and rated false news as more false than they rated true news as true (by 0.28pts).

In Wave 2, in the treatment, participants showed a small skepticism bias, and rated false news as more false than they rated true news as true (by 0.14pts).

For the measure of truth-bias we follow: <https://osf.io/n9h4y/>”

Reviewer 2: 6. A cynical reader (not myself, for the record) would say “so what?” to these findings. That is, if you ask people to follow (good) news compared to non-news, no wonder they had positive effects on news-related outcomes! The longitudinal, multi-site, multi-platform design — that occurred in the wild — counterbalances this cynical view. Therefore, I'd like to see the

authors articulate the impact and value of this work more in the Discussion. Tell the readers why this work is practically and theoretically compelling beyond the obvious methodological improvements on prior work.

Authors: Following this good comment, we have greatly extended the discussion, as well as the theoretical and practical implications of our findings (see comments above). In the Introduction, we also better situate our findings in the current literature on social media news, which is, in general, relatively pessimistic regarding the benefits of social media news.

“Researchers worry that news on social media has negative effects, by increasing information overload and political polarization, or by increasing feeling of knowing – without actually learning anything.” p.2

“These two core findings, together, suggest that improving exposure to verified information from news media could be more effective than reducing exposure to low-quality information from junk news. The increased belief in true news and trust in news and journalists is important given the evidence that the rejection of true news may be a bigger problem than the acceptance of false news. For instance, people are skeptical of true news \cite{pfanderAltay2023meta}, distrust news media in general, including reliable news outlets \cite{newman2023digital}, and trust mainstream news outlets much less than fact-checkers do \cite{pennycook2019fighting}. This suggests that improving exposure to reliable information may be more beneficial \cite{askari2024incentivizing, altay2024tips} than reducing exposure to misinformation, which---after all---represents a small portion of the news people see online (\cite{acerbi2022research}).” p.5

“Although this project offers new optimistic evidence, we encourage future research to examine the generalizability of our findings. For one, it is unclear whether similar effects would emerge on platforms with different features or uses. In our case, we do not detect statistically significant differences between platforms, but we lack statistical power to detect small differences. Yet, there are reasons to expect that platforms with different algorithms and affordances would yield distinct results. For instance, if participants had followed news accounts on TikTok, they may never have been exposed to content from these accounts because users mostly use the "For you" feed which weakly prioritizes followed accounts. Similarly on YouTube, where the algorithms direct users away from news \cite{huang2024auditing}, subscribing to news channels may not increase news exposure in any pronounced ways (but tweaking the algorithm itself does generate these increases \cite{yu2023nudging}). However, holding the quantity and quality of news constant, exposure to news on TikTok, YouTube or other platforms should also yield the benefits documented here.” p.6

“The beneficial effects detected may be explained through cognitive processing theories \cite{cacioppo1986central}: the concise, visually-driven nature of short-form news on platforms like Instagram and WhatsApp may enhance information retention by reducing cognitive overload, while the interactive features of these platforms could foster deeper engagement. Unlike traditional, longer formats, the processing of which requires deeper, more reflective processing (per the Elaboration Likelihood Model \cite{petty2011elaboration}, short-form news may rely more on peripheral cues like visuals or source credibility. This could mean that the cognitive and emotional pathways through which these formats influence knowledge retention, trust, and content

discernment differ substantially from those activated by traditional media. We encourage scholars to explore these under-studied mechanisms.” p.6

Reviewer 2: 7. One of the recommendations offered or implied by the authors is that following good news may help people achieve democratic ideals. Who defines what is “good” here? I am writing from the US, and we are quite divided in America over the media. To those who are left-leaning and liberal, the NYTimes and MSNBC are often perceived as “good,” and Fox News, Breitbart, etc. are “bad.” To those who are right-leaning and conservative, the opposite is true. Goodness or badness is therefore (unfortunately) subjective, and not objective. There are media bias scores that might indicate goodness or badness (<https://adfontesmedia.com/interactive-media-bias-chart/>), but these perceptions are in the eye of the news consumer. How do the authors contend with the idea that you simply will not get a very right-leaning and conservative person on social media to follow the NYTimes? See recent Pew data: <https://www.pewresearch.org/journalism/2014/10/21/political-polarization-media-habits/>. Of course, the experimental results in this work had random assignment so any ideology effect is likely accounted for in the design. But, practically speaking, can this study — plus the ethos and design behind it — actually inform or intervene in the ways the authors idealize? This is curious to me, and I am skeptical. Perhaps this would work in other countries that are less divided than the US.

Authors: This is a very important point. First, we would like to push back on the idea that news quality is purely subjective: ratings of news outlets quality strongly correlates with laypeople perceptions in the US and in Europe (Pennycook & Rand, 2019; Schulz, Fletcher & Popescu, 2020). Similarly, despite partisan differences and polarization, people discern true from false news, even when the news is political and partisan (<https://osf.io/n9h4y/>). That being said, you are right that partisan affiliation clearly influences perceptions of news outlets and of the accuracy of news stories.

You are also right that this question is harder to answer in countries like the US given the fragmentation of the news ecosystem and political polarization, than in countries like France, Germany or the UK, where there is a strong public broadcast service that people can trust and turn to for news. Perhaps in the US it could work with some international news outlets that are not seen as clearly partisan like the BBC or Reuters? In sum, we believe that there are enough high quality news outlets that post in English for our approach to work in a country like the US. However, we now recognize in the discussion that our approach is premised on the existence of reliable news outlets posting on social media, which may not be the case on some platforms, in some countries/languages.

“Studies should replicate our findings in different countries and different times in electoral politics. Our experiment was run in two multi-party systems with strong public-service broadcasting and during relatively contentious times (with EU elections, Israel's attacks on Gaza, and the Russian attack on Ukraine). The effects may be different in other periods and contexts. Further, our approach is premised on the presence of reliable news outlets posting on social media, yet, in some countries or on some platforms, such outlets may be scarce” p.6

Reviewer 2: 8. Can the authors elaborate on the ethics approval of this study and the ethical considerations of presenting people with fake/false news? Yes, this is a research study and presumably participants may have known that they would be presented with some true or untrue content. But, perhaps people believed what they saw. This presents a major ethical issue if

proper debriefing and discussion of the experimental materials were not performed (e.g., people communicating to their friends, after the study, about what a global crisis). Please elaborate and write about this at length in the manuscript.

Authors: Thank you for this good question. First, we would like to stress that participants were not deceived in the experiment: we did not expose them to “fake news” or misinformation as treatment, nor did we present false statements as true. Instead, our treatment consisted of following verified and “quality” news organizations (per the point above). Simply, the questionnaires exposed participants to true and false statements, asking the participants to indicate whether they have heard the statements before and whether they believed the statements to be accurate or not. In other words, the false statements were merely part of the questionnaires, not the treatment.

In addition, this procedure -- or even a more “aggressive” one that directly exposes participants to false information as part of the experimental treatment! -- is very common in the scientific literature on misinformation. In the past four years, hundreds of studies have exposed online participants to true and false news (<https://osf.io/n9h4y/>). To study misinformation and test interventions against it, it is a necessity to expose participants to news or claims that are not accurate. And while there is evidence that many studies in the field do not properly debrief participants or forget to report it (<https://doi.org/10.1027/1016-9040/a000491>), our participants were debriefed.

We debriefed all participants by email after the end of wave 2. We could not debrief participants at the end of wave 1, as it would have impeded our treatment. Debriefing participants by email allowed us to reach not only participants who finished both waves but also participants who dropped out and, e.g., only finished wave 1. Please find below an anonymized version and English translation of the debriefing email (French participants received a French version while German participants received a German version, some debunking links were also specific to German/French participants).

“You participated in a study in which you were asked to follow accounts on WhatsApp or Instagram. Please find some important information about the study below.

The purpose of the study was to measure the effect of following the news on social media. If you were asked to follow non-news accounts, it means that you were in the Control Condition. We measured the effect of following news accounts (versus not following any) on various outcomes including political knowledge, awareness of true and false news, as well as belief in true and false news. The items in the survey were not personally targeted towards you. In the survey, you were asked to rate four false statements and we would like to set the record straight: (i) it is true that the Israeli army killed more than 30'000 Palestinians, (ii) the Palestinians victims are not ‘crisis actors’ – they are humans, often women and children; (iii) NATO never promised Russia not to expand after the Cold War, finally (iv) Ukraine did not commit a genocide in the Donbass. If you would like to know more about these claims, please visit these websites:

(i)<https://www.npr.org/2024/02/29/1234159514/gaza-death-toll-30000-palestinians-israel-amas-war> (ii)<https://www.npr.org/2023/11/27/1214451419/civilian-deaths-are-being-dismissed-as-crisis-actors-in-gaza-and-israel>

French:(iii)<https://euvsdisinfo.eu/fr/desinformation-sur-le-conflit-actuel-entre-la-russie-et-lukraine-sept-mythes-deboulonnes/> French:(iv)<https://euvsdisinfo.eu/fr/le-manuel-strategique-du-kremlin-inventer-un-pretexte-pour-envahir-lukraine-plus-de-mythes/>
German:(iii)<https://euvsdisinfo.eu/de/desinformation-ueber-den-aktuellen-russland-ukraine-konflikt-sieben-mythen-entlarvt/> German:(iv)<https://euvsdisinfo.eu/de/das-regelbuch-des-kremls-schaffung-von-vorwaenden-fuer-den-einmarsch-in-die-ukraine-mehr-mythen/>

The main researchers conducting this study are HIDDEN (HIDDEN@HIDDEN) and HIDDEN (HIDDEN@HIDDEN), both HIDDEN. If you have questions about this study, or would like to know the results of the study, you may contact them. If you have concerns regarding your rights as a research participant in this study, you may contact the Ethics Committee.

Best,
HIDDEN & HIDDEN

Contact of the ethics committee: Prof. HIDDEN President of the Ethics Committee
Institute of Psychology chair.ethics.committee@phil.HIDDEN.HIDDEN HIDDEN

We've added information about the debriefing procedure and information in the manuscript, together with existing information about the ethical approval of the study. The full text of the recruitment materials, the consent form, and the debriefing are now available on OSF.

Furthermore, please note that overall, participants were overwhelmingly exposed to true statements, with 14 true statements (news + claims) vs 4 false statements in wave 2. Thus, any negative effect of exposure to the false statement was largely counterbalanced by the positive effect of exposure to true statements.

We also want to stress that on average, participants rated the false claims as false. The average perceived accuracy of the false claims in Wave 1 is 3.73 – “3” corresponds to “Mostly false” and “4” to “A little false”. With 81% of participants having a mean lower than 5 (“A little true”), 94% of participants having a mean lower than 6 (“Somewhat true”) and 98% of participants having a mean lower than 7 (“Very true”) - suggesting that very few participants actually fell for the false claims. Moreover, given that the treatment increased belief in true claims, as well as current affairs knowledge and trust in reliable sources of information, we believe that any potential effect of exposure to the false claims was counterbalanced.

We trust that this response as well as all the additions to the final manuscript and the Appendix well address this important comment.

Reviewer 2: Altogether, I find the work interesting, but I need more from the authors. I look forward to seeing them take up the gauntlet in a possible revision.

Authors: We again thank you for all the positive words on the manuscript and the great suggestions and detailed feedback. We hope that these changes fully address all the comments.

Reviewer 3

Reviewer 3: This paper addresses the ongoing debate on whether people can learn from or via social media about political events and actors. Although, the paper is not novel in terms of concept or theory, it uses a very innovative approach in terms of data and methods. In contrast to most studies on this topic it uses a field experiment to expose individuals to news on different social media platforms. By incentivizing social media users in France and Germany to follow quality/mainstream news outlets on Instagram and WhatsApp for two weeks. In this way the study does not try to correct the 'bad' information out there, but focuses on 'promoting' reliable information about recent political events. The use of compliance measures in the study is innovative and really well done.

Authors: Thank you for noting that our manuscript uses a very innovative approach, and that the use of compliance measures is innovative and well done. Thank you also for your useful comments and queries. We address each of these issues below.

Reviewer 3: Overall, the data approach seems valid to me, gathering high quality data, and also the presentation of the empirical findings is clear and solid. I believe the authors are a bit too 'positive' in their interpretation of the data. Not all results are significant (depending on platform and country), but more importantly the effects seems rather small (taking into account the relative large N). A bit more attention to the effect sizes would be insightful.

Authors: You are absolutely right that the effect sizes of our field experiment are small. We try to be very clear about this in the manuscript. However, we would like to note that effect sizes of such field experiments are small in general because the content of the treatment (news in that case) is not forced on participants, and because compliance is never perfect. We would also like to note that the main analyses we pre-registered are aggregated across countries and platforms for power reasons. We are not confident in the reliability of country-platform specific results across country/platforms, as we do not have enough power to reliably detect small country/platforms specific effects and differences.

We now emphasize clearly the null effects and the small effect sizes in the discussion:

“For these reasons, it is important to keep in mind that, despite being statistically significant, the treatment effects are small. We also found numerous non-significant effects, such as awareness of and belief in false news stories, the feeling of being informed, interest in news and politics, political efficacy, and feelings towards Ukraine, Russia, Palestine, Israel, Europe, and specific political parties. These null effects are in line with past work showing that attitudes are very resistant to change \cite{nyhan2023like, guess2023algorithms, yu2023nudging}.” p.5

Reviewer 3: In the method section (4.9) the authors mention they control for age, gender, education, and past vote. But what about controlling for media use? I guess the effects are most outspoken for those that have a low news diet? (and perhaps a ceiling effect for news junkies?). This seems more important to me than controlling for past vote. (News use is mentioned as a moderator in the preregistration of W1)

The Preregistration is extensive and clear. All the deviations from the pre-registered protocol are well documented and make sense.

Authors: As pre-registered, we investigated the moderating effect of media use (in wave 1) on treatment effect and found no significant moderating effects. We report these on OSF because there are too many statistical tests to be reported in the manuscript, here are the results for media use:

To fully address this comment, we also re-run the main models (Fig1) while controlling for news consumption in wave 1, and found that the main effects hold:

We now report this figure in Appendix K.

Reviewer 3: In the conclusion the authors make a convincing claim that incentivizing social media users to follow news on Instagram and WhatsApp can lead to knowledge gains, and other positive democratic outcomes. However, without disputing these ‘hopeful’ results, the authors might be a bit more critical. The paper stresses the possibility of a virtuous circle between news consumption on social media and algorithmic suggestions for more news, but nothing is mentioned about a potentially more vicious circle where people decide to follow more alternative/partisan or low quality news outlets. And what if people decide to no longer subscribe to a newspaper and prefer the short/free news updates on WhatsApp? Again, I think the potential positive effects are real, but a more nuanced discussion of these effects seems warranted. Next, I believe the discussion could be more relevant for scholars working on this topic if the authors would interact more with previous studies. Some studies (cited in this study) have speculated about the non-findings of social media on political knowledge and suggested the lack of actual news content on social media as one of the potential explanations (eg. Boukes; Van Erkel & Van Aelst). This study provides actual proof that this is one of the main drivers. People perceive that they have seen news stories on social media, but in reality they often have not (or not enough).

Authors: We agree that the possibility of vicious circles is worth mentioning. We now discuss it in the discussion:

“On the downside, our approach could have negative substitution effects by increasing social media news consumption while reducing the consumption of higher-quality news. Yet, such substitution effects are rare (Nyhan 2010 corrections, guess 2023 algorithms) and incompatible with our treatment effects.” discussion

We now engage more with past mechanisms mentioned to explain the null or negative effects of social media on political knowledge:

“Researchers worry that news on social media has negative effects, by increasing information overload and political polarization, or by increasing feeling of knowing – without actually learning anything \citep{lee2024self, van2021don}.”

“We note that these findings are more in line with past deactivation studies, which show that de-activating Facebook reduces knowledge \citep{arceneaux2024facebook, allcott2020welfare, asimovic2021testing}, than with other, mostly cross-sectional, work, showing that platforms like WhatsApp, Instagram, or Facebook, have null or negative effects on political knowledge \citep{amsalem2023people, yu2023nudging}.”

“For instance, \citeauthor{askari2024incentivizing} created bots that contextually replied to users tweeting about non-political topics and encouraged the users to follow verified and balanced news accounts. This intervention slightly enhanced the following and liking of news accounts. In a YouTube field experiment, \citep{yu2023nudging} found that nudging the algorithm by playing videos from news channels in the background increases recommendations to and consumption of news over-time and promotes more diverse news diets. Such positive incentivizations align with users’ desire for accurate information and educational content, instead of divisive, hateful, or false content \citep{rathje2023people} p.2

Reviewer 3:Additional points for Improvement:

WhatsApp is treated as a social medium in this paper. Most studies label it as a (private) messenger app. Please clarify why WhatsApp is considered a social media platform.

Authors: This is a tricky question. While it is true that we treat WhatsApp as a social media platform, we define it as “a messaging platform with almost three billion unique active users worldwide”. Yet, WhatsApp is more than a messaging app: users can share stories with their contact (via the ‘status feature’, which is similar to stories on Instagram or Facebook), they can share videos and images, join large groups or communities, and follow the news with the new ‘News tab’. We believe that these features fall within many definitions of social media, including Mirriam Webster's definition: forms of electronic communication (such as websites for social networking and microblogging) through which users create online communities to share information, ideas, personal messages, and other content (such as videos). We have changed our definition of WhatsApp to make it clearer that it’s primarily a messaging app with social media features:

“WhatsApp is a messaging platform with numerous social media features (such as stories) and almost three billion unique active users worldwide” p.3

In response to Reviewer 1 we have also extended the discussions on affordances offered by the platforms.

“We focus on Instagram and WhatsApp, platforms with different affordances and distinct usage patterns. Instagram is a well-established image and video-based social media platform with over two billion monthly active users worldwide as of early 2024. WhatsApp

is a messaging platform with numerous social media features (such as stories) and almost three billion unique active users worldwide \citep{nobre2022hierarchical}. Each offers distinct opportunities for incidental news exposure: Instagram provides most opportunities, as news content is displayed on the same page as friends' posts and stories, while WhatsApp offers fewer, as users must actively open the news tab to access news." p.3

"Although this project offers new optimistic evidence, we encourage future research to examine the generalizability of our findings. For one, it is unclear whether similar effects would emerge on platforms with different features or uses. In our case, we do not detect statistically significant differences between platforms, but we lack statistical power to detect small differences. Yet, there are reasons to expect that platforms with different algorithms and affordances would yield distinct results. For instance, if participants had followed news accounts on TikTok, they may never have been exposed to content from these accounts because users mostly use the "For you" feed which weakly prioritizes followed accounts. Similarly on YouTube, where the algorithms direct users away from news \citep{huang2024auditing}, subscribing to news channels may not increase news exposure in any pronounced ways (but tweaking the algorithm itself does generate these increases \citep{yu2023nudging}). However, holding the quantity and quality of news constant, exposure to news on TikTok, YouTube or other platforms should also yield the benefits documented here." p.6

Reviewer 3: I find the term 'news knowledge' a bit confusing. In my view the authors measure political knowledge, or what in the literature is labelled as 'surveillance political knowledge' (or current affairs knowledge). It is about recent political events and actors that were featured in news coverage. The authors also sometimes speak about political knowledge, so it would be more consistent to use this term (note: in the preregistration the term surveillance knowledge is used). In that regard, it might be useful to document the knowledge questions a bit more in the appendix. What was the number of correct answers in wave 1 and wave2? To what extent were the answers to this questions mentioned in the news outlets used in the experiment (in the two weeks period)?. In other words, how much opportunities were there to learn? It might be that the question about the current Minister of National Education and Youth? (W1 and W2 FRANCE) was not given news attention, so you would not expect people to improve their knowledge (and in that case the term 'news knowledge' is simply incorrect).

Authors: You are absolutely right that the technical term for 'news knowledge' is 'surveillance political knowledge' or 'current affairs knowledge'. We opted for 'news knowledge' because we anticipated that readers of NHB may be confused by the technical terms. In any case, we agree and changed 'news knowledge' to 'current affairs knowledge' – as it may be easier to interpret than the term 'surveillance political knowledge', which may sound weird to readers outside of the field.

In Appendix F, we have added information about the descriptives for current affairs knowledge, as well as awareness of true and false claims (note that in that Appendix there is already a descriptive table for belief in true and false claims).

"These descriptives are the predicted values from the statistical models of the main analyses (i.e., the intent-to-treat analyses adjusting for demographics, etc.).

News knowledge

In Wave 1, participants in the control and the treatment had respectively 1.47 and 1.46 correct responses – the minimum being 0 and the maximum 3.

In Wave 2, participants in the control and the treatment had respectively 2.87 and 2.97 correct responses – the minimum being 0 and the maximum 7.

Awareness of true claims

In Wave 1, participants in the control and the treatment reported being aware of, respectively, 1.15 and 1.12 true claims – the minimum being 0 and the maximum 2.

In Wave 2, participants in the control and the treatment reported being aware of, respectively, 3.34 and 3.45 true claims – the minimum being 0 and the maximum 7.

Awareness of false claims

In Wave 1, participants in the control and the treatment reported being aware of, respectively, 0.32 and 0.31 false claims – the minimum being 0 and the maximum 2.

In Wave 2, participants in the control and the treatment reported being aware of, respectively, 1.01 and 1.03 false claims – the minimum being 0 and the maximum 4.”

We do not know precisely the extent to which the answers to the questions were actually in the news coverage of the accounts participants were asked to follow in the treatment. However, because we selected relatively major news events covered by most news outlets, and because we tried to manually check whether the specific news outlets participants were asked to follow covered these events, we are confident that most of the current affairs knowledge questions, as well as most of the true claims, were covered by the news outlets during the time of the experiment.

We are also confident that the news outlets the participants were incentivized to follow did not cover the false claims. These news outlets do very little fact-checking or debunking, and are known to be verified, factual, and credible legacy media.

Regarding the specific example you mentioned about the French Minister of National Education and Youth, there is no doubt that it was covered in French news because Nicole Belloubet took over one month before the start of Wave 1 and because the minister before her, Amélie Oudéa-Castéra, lasted for less than a month at this position and it created a small media scandal, e.g., <https://www.france24.com/en/france/20240116-private-school-faux-pas-spells-trouble-for-france-s-new-education-minister>
https://www.lemonde.fr/en/education/article/2024/01/16/amelie-oudea-castera-the-education-minister-accused-of-fleeing-public-schools_6436588_104.html

<https://www.bloomberg.com/news/articles/2024-02-08/macron-replaces-education-minister-who-sparked-teacher-anger?embedded-checkout=true>

Reviewer 3: Overall, this is a truly innovative study that is well executed and provides relevant results. My suggestions for improvement are rather minor.

Authors: We again thank you for all the positive and constructive feedback. We hope that these changes fully address all your great suggestions.

In conclusion, we thank the editor and the reviewers for your close reading and feedback and hope that you will see the paper as ready for publication in *Nature Human Behavior*.

Dear [REDACTED]

Please find attached all the files necessary for the publication of the manuscript.

We have addressed the two remaining comments from the reviewers by removing the sentence in the abstract that Reviewer 3 disliked, and by precisising that “Israel's attacks on Gaza” occurred “following the October 7th attacks by Hamas” (per Reviewer 2).

We would like to participate in transparent Peer Review.

Best,

Sacha, Emma, and Magdalena